# A decade of continuous Rockall Trough transport observations using moorings and gliders

Kristin Burmeister[1], Sam C. Jones[1], Neil J. Fraser[1], Alan D. Fox[1], Stuart A. Cunningham[1], Lewis A. Drysdale[1], Mark E. Inall[1], Tiago S. Dotto[3], and N. Penny Holliday[3]

[1]Scottish Association for Marine Science, Oban, UK
[3]National Oceanography Centre, Southampton, UK

**Correspondence:** Kristin Burmeister (kristin.burmeister@sams.ac.uk)

**Abstract.** The Rockall Trough, northwest of Scotland and Ireland, is a key conduit for the North Atlantic Current (NAC) and European Slope Current (ESC) transporting heat and salt toward the Nordic Seas and Arctic Ocean while mediating exchanges between the open ocean and the European shelf. We present a decade-long record of Rockall Trough circulation from the Ellett Array providing the first continuous estimates of heat and freshwater transport between 2014 and 2024. We develop a methodology that combines the high spatial resolution of gliders with the high temporal resolution of moorings and ocean reanalysis output producing continuous eastern boundary velocity fields of the ESC for integration into the full Rockall Trough transport product. This approach improves the mean structure of the ESC, capturing the southward undercurrent previously unresolved and enhancing the ability to reproduce extreme, likely mesoscale, transport events. The Rockall Trough transport is dominated by the NAC flowing through the mid basin, exhibiting multi-year variability consistent with changes in the subpolar gyre and the mid-2010s cold freshwater anomaly. The ESC acts as a secondary driver, is not correlated with the NAC and is influenced by along-slope wind stress. Since 2022, warmer and saltier conditions, amplified by the 2023 extreme North Atlantic marine heatwave, have strengthened northward volume, heat, and salt transport through the Ellett Array. Our results highlight the value of sustained glider-based boundary current observations for Atlantic climate monitoring and demonstrate that the combined mooring–glider framework provides a robust and transferable approach for long-term ocean transport monitoring.

**Todo list**

# 1 Introduction

The Rockall Trough, located just off the continental shelf northwest of Scotland and Ireland, is a vital passageway for currents that shape Earth's climate—carrying heat and salt toward the Nordic Seas and Arctic Ocean and enabling exchanges between the open ocean and the European shelf. This occurs through the eastern most branch of the northward flowing North Atlantic
Current (NAC) in the center of the Rockall Trough and the European Slope Current (ESC), which flows northward along the Rockall Trough's eastern boundary. Together, these currents exert a strong downstream influence on physical and biogeochemical conditions in the Northwest European Shelf, North Sea, Nordic Seas, and Arctic Ocean (Berx et al., 2013; Marsh et al., 2017; Porter et al., 2018; Johnson et al., 2024).

The NAC transports heat from the Gulf Stream toward the Nordic Seas and divides into several branches that flow through the
25 Iceland Basin (Dotto et al., 2025), across the Rockall Plateau (Houpert et al., 2018) and through the Rockall Trough (Houpert et al., 2020; Fraser et al., 2022). Recent results indicate that the NAC flow through the Rockall Trough plays a primary role in the variability of the subpolar overturning circulation (Fu et al., 2025), while buoyancy loss to the atmosphere from NAC branches farther west represents the largest water mass transformation signal in the subpolar gyre (Jones et al., 2023).

The ESC is smaller in volume transport than the NAC but exerts a strong influence on hydrographic conditions of the
30 Northwest European Shelf (Porter et al., 2018; Jones et al., 2020) and of the North Sea (Marsh et al., 2017). It carries warm and saline Eastern North Atlantic Water along the shelf break, facilitating exchanges of these water masses with the shelf. Large-scale variability in the subpolar gyre can affect the source waters and the strength of the ECS by altering the currents that feed into it (Clark et al., 2022; Daly et al., 2024).

While the NAC and ESC form the easternmost branches of the North Atlantic subpolar gyre, their local variability within the
35 Rockall Trough is governed by distinct physical processes. Variability of the NAC within the Rockall Trough can be explained by the thermal wind relation (Fox et al., 2022; Holliday et al., 2020), with changes in the local zonal density gradient being mainly temperature-driven (Fraser et al., 2022). In contrast, the ESC is primarily influenced by the large-scale meridional density gradient and surface wind stress (Huthnance, 1984; Huthnance et al., 2022; Marsh et al., 2017). Shifts in the size and position of the subpolar gyre affect both the source waters feeding the Rockall Trough circulation and the relative strength of
40 the Rockall Trough NAC branch, often in opposition to the NAC branches further west (e.g., Holliday et al., 2020; Foukal and Lozier, 2017; Häkkinen and Rhines, 2004). These differences are reflected in observed transport time series of different NAC branches (Dotto et al., 2025; Fraser et al., 2022). Understanding long-term, climate-relevant changes in poleward heat and freshwater transport as well as variations in physical and biogeochemical fluxes onto the European Shelf, therefore requires continued monitoring of both the NAC and ESC. The Rockall Trough provides a natural bottleneck in the North Atlantic
circulation for achieving this goal.

Although the NAC has been well constrained through longstanding efforts using mooring and hydrographic observations in the Rockall Trough (e.g. Ellett et al., 1986; Holliday and Cunningham, 2013), only the recent use of sustained glider missions

has enabled consistent monitoring of the ESC (Fraser et al., 2022). Regular hydrographic surveys on the Extended Ellet Line, a section across the northern Rockall Trough, began in 1975, with a southern Rockall Trough section established in 2006 (Daly et al., 2024). Since 2014, transports across the Extended Ellet Line in the Rockall Trough have been continuously monitored using a dynamic height mooring array, called the Ellett Array (Figure 1) deployed as part of the larger OSNAP (Overturning in the Subpolar North Atlantic) programme (Houpert et al., 2020; Lozier et al., 2019). While the Ellett Array accurately constrains the volume transport of the NAC in the Rockall Trough mid basin, difficulties have arisen at the Rockall Trough boundaries where narrow jets like the ESC over steep topography exhibit high spatiotemporal variability and are less suited to moored observations (Houpert et al., 2020). Over recent decades, several studies have made moored observations of the ESC velocity (summarised in Diabaté et al., 2025), however these have generally been short-term process studies rather than long-term climate monitoring efforts. A first milestone towards long-term monitoring of the ESC was reached with the introduction of repeated glider observations at the eastern boundary of the Ellett Array in 2020 (Fraser et al., 2022). Because glider data are heterogeneous in space and time, Rockall Trough transport derived from moorings and ESC transport derived from glider observations are currently treated as separate data products.

Repeated glider surveys have substantially improved our understanding of the ESC. Although spatial scatter has been mitigated through gridding methods, temporal intermittency continues to limit long-term analyses. Fraser et al. (2022) developed an along-isobath transformation to map spatially scattered glider data onto the Ellett Array section grid, enabling hydrography and velocity fields from each transect to be analysed on a consistent gridded section while preserving volume and property transports. This approach has yielded unprecedented insight into the strength, structure, and seasonality of the ESC and provides a directly observed transport estimate that can validate mooring-based extrapolations of the eastern wedge of the Ellett Array section (Figure 1). However, although Fraser et al. (2022) addresses spatial variability, the gridded glider transects remain temporally sparse and intermittent. Time gaps between glider transects can vary from one day up to several months, so results concerning ESC variability must be interpreted with caution. To be fully integrated into climate monitoring, glider observations must ultimately provide a continuous, regularly sampled transport time series.

In this study we extended the Rockall Trough transport product of previous studies (Houpert et al., 2020; Fraser et al., 2022) to a full decade of observed volume transports and, for the first time, present associated heat and freshwater transports. We additionally construct a continuous, regularly sampled Rockall Trough eastern wedge velocity product by combining the high temporal resolution of the moored observations with the high spatial resolution of the glider sections. This approach improves the spatial representation of the ESC within the total Rockall Trough transport time series, providing a better representation of mean structure and mesoscale variability compared to the previous method (Houpert et al., 2020; Fraser et al., 2022). The method developed here is directly applicable to the broader challenge of ocean transport monitoring using heterogeneous arrays of mooring and glider platforms.

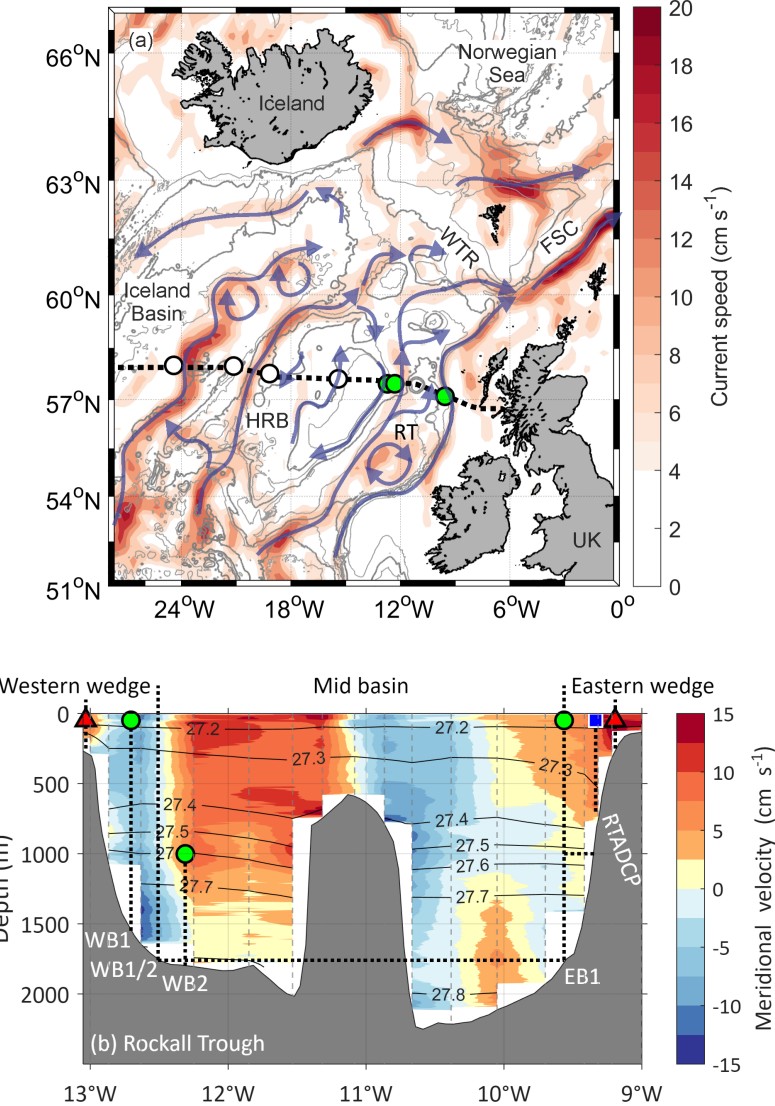

**Figure 1.** a) Mean surface current speed (1993–2023) from satellite data with schematic North Atlantic upper ocean circulation (blue arrows) and Ellett Array mooring positions (white and green circles). Full array shown for context; this study focuses on Rockall Trough moorings (green circles). b) Mean meridional velocity (shading, positive northward) from 17 repeated lowered ADCP sections along the Extended Ellet Line (1996–2017). Dark grey lines show associated mean potential density (kg m$^{-3}$), vertical grey dashed lines mark the lowered ADCP/CTD stations. Vertical dotted lines with green circles mark mooring locations (WB1, WB2, EB1), with red triangles mark section endpoints (13°W, 9.2°W), with blue rectangle marks RTADCP (2014-2015). Vertical dotted line labeled WB1/2 is partition point of western wedge and mid basin, EB1 is partition point of mid basin and eastern wedge. Horizontal dotted line between WB1/2 and EB1 marks maximum depth for mid basin transport estimation. Horizontal dashed line east of EB1 marks maximum depth of glider monitoring upper 1000 m of eastern wedge since 2020. Acronyms: Rockall Trough = Rockall Trough, HRB = Hatton-Rockall Basin, WTR = Wyville Thomson Ridge, FSC = Faroe-Shetland Channel, UK = United Kingdom, ADCP = Acoustic Doppler Current Profiler.

## 2 Data and methods

 ## 2.1 Moored observations

Moored hydrographic and velocity observations are obtained from three hydrographic subsurface moorings (WB1, WB2, EB1) at either side of the Rockall Trough from 2014 to 2024 (Houpert et al., 2020; Fraser et al., 2022). All moorings are equipped with Sea-Bird SBE37MP Conductivity-Temperature-Depth sensors (CTDs) and Nortek Aquadopp single point current meters distributed throughout the water column (Figure 2). Figure 1 shows the location and depth of the different moorings: EB1 (57.1°N, 9.6°W) and WB1 (57.5°N, 12.7°W) are full-depth moorings extending from the seabed (EB1 at 1800 m, WB1 at 1600 m) to about 50 m. WB2 (57.5°N, 12.3°W) is located downslope of WB1 at 1800 m water depth to extend the western boundary observation to the same depth as EB1. Following Fraser et al. (2022), we create a virtual mooring, WB1/2, positioned midway between WB1 and WB2 at 12.5°W. It combines temperature, salinity, and velocity observations from WB1 above 1600 m and from WB2 below 1600 m. The near-flat isopycnals between WB1 and WB2 (Figure 1) justify directly concatenating their hydrographic data. Similarly, velocities above 1600 m are horizontally uniform between WB1 and WB2. Below 1600 m, flow between WB1/2 and WB2 is weak and contributes negligibly to transport (Fraser et al., 2022) justifying the use of WB2 current meter data here.

Temperature and salinity data are corrected for sensor drift pre- and post-deployment using in situ CTD profiles. Over a two year deployment period, the accuracy of the calibrated moored salinity, temperature and pressure data are estimated to be 0.003, 0.002°C and 1dbar, respectively (McCarthy et al., 2015). Temperature and salinity data are then linearly interpolated on to a regular 2-hour time grid and de-spiked. Horizontal velocity measurements are corrected for speed of sound and magnetic deviation.

The corrected hydrographic and velocity data are de-tided using a 48-hr lowpass filter, then linearly interpolated on a 20-dbar vertical grid and a 12-hr temporal grid. The gridded temperature and salinity fields were extrapolated to the surface by repeating the uppermost observed values vertically at each time step. The gridded velocity fields are linearly extrapolated to the surface, despiked and any gaps are then interpolated horizontally over the time dimension (Houpert et al., 2020; Fraser et al., 2022).

The data return of the moored hydrographic and velocity instruments are generally very high (see black solid lines marking pressure records of each instrument in Figure 2). At EB1, 78% of all CTD data and 89% of all current meter data were recovered. At WB1 and WB2, 85% of CTD data were recovered, along with over 99% of current meter data at WB1 and 97% at WB2. A summary of the different gap filling methods applied to the gridded mooring data over the years is given in the supplementary information (Section S1).

A bottom-mounted Acoustic Doppler Current Profiler (ADCP) was deployed in the ESC core in 2014 but only the first 8 months of data could be recovered. In the following we will refer to instrument as RTADCP. Attempts to recover data from later ADCP deployments failed because the instruments were severely damaged, presumably by fishing activities (Houpert et al., 2020). Consequently, the Rockall Trough transport product relies on ocean reanalysis output, bias-corrected using the

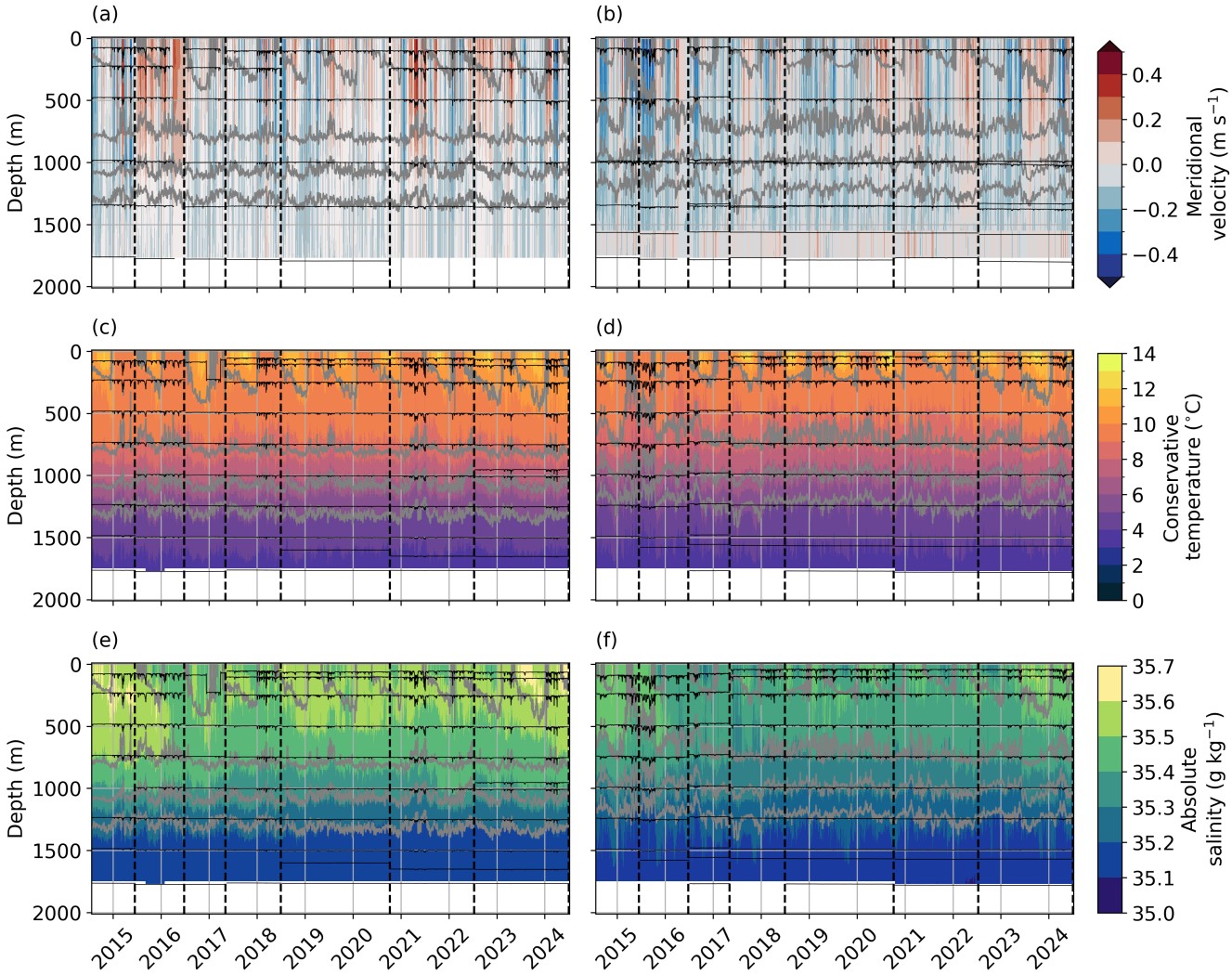

**Figure 2.** Gridded fields of moored meridional velocity (a-b), conservative temperature (c-d) and absolute salinity (e-f) of the concatenated mooring WB1 and WB2 (a,c,e) and EB1 (b,d,f). Black lines mark the depth derived from the pressure time series for the individual current meters (a-b) and CTD sensors (c-f). Grey contour lines show potential density values at 27.2, 27.4, 27.6 and 27.7 kg m$^{-3}$ (top to bottom). Vertical dashed black lines indicate the individual servicing times for the moorings.

available eight-month RTADCP time series, to reconstruct the northward flow of the ESC following Houpert et al. (2020) and Fraser et al. (2022).

## 2.2 Glider observations

Seven consecutive glider missions occupied the eastern Rockall Trough boundary region during the periods April–August 2020, October 2020–January 2021, April–May 2021, October 2021–February 2022, April 2022, July 2022-September 2022, November 2022-February 2023. The instruments completed a total of 166 repeat transects between the approximate EB1 mooring location and the 200 m isobath to the east (Figure 1b). The gliders measure temperature, salinity and pressure between the surface and the seabed up to a maximum depth of 1000 m (Fraser et al., 2022).

All gliders are equipped with SBE41 CTD senors, which according to the manufacturer have an accuracy of 0.0035, 2dbar and 0.002°C and a typical stability of 0.0011 per year, 0.8 dbar per year and 0.0002°C per year for salinity, pressure and temperature, respectively. Glider CTD sensors and compasses were calibrated in the laboratory before each mission, and an in-water compass calibration was performed at the start of each deployment. Gliders navigate with a magnetic compass under water and obtain a GPS position fix each time they surface. Depth-Averaged Current (DAC) can be derived based on any deflection off course between two consecutive GPS fixes (Fraser et al., 2022). Coefficients of the gliders hydrodynamic flight model (Eriksen et al., 2001) were re-calculated after each mission by selecting a subset of dives and iteratively comparing the vertical velocity predicted by the flight model against the vertical velocity implied by the pressure sensor. New estimates of horizontal and vertical velocities were then used to correct for thermal inertia effects on the unpumped CTD sensor. The corrected data profiles were merged to a single data file using GliderTools (Gregor et al., 2019). The data were then despiked using a rolling median filter.

Fraser et al. (2022) developed a method to estimate volume transport in the upper 1000 m of the eastern wedge using glider observations by projecting irregular transects onto a common section. Glider trajectories are influenced by ocean currents, resulting in irregular paths and transects that differ in location and cross-sectional area (Figure S1). To reconcile these inconsistencies, all transects are projected onto the eastern wedge subsection, spanning from mooring EB1 at 9.6°W to 9.2°W along 57.1°N (Figure 1b), using an along-isobath transformation. This common section is oriented approximately perpendicular to the isobaths. The approach assumes that ESC streamlines follow isobaths and that tracers are conserved along isobaths within the glider survey region. A summary of the method is provided below, with full details in the supplementary information (Section S2).

For temperature and salinity, in situ measurements along individual glider transects are allocated to coordinates along the common section that share the same isobath. DAC observations require additional steps: (i) The DAC component perpendicular to the glider path is assigned to the same isobath on the common section and rotated to be perpendicular to that section. (ii) This value is scaled by the ratio of seabed slope along the glider path to that on the common section, ensuring transport between isobaths is conserved. Transformed data are then interpolated onto a two-dimensional grid with a horizontal resolution of $dx \sim 250$ m and vertical resolution of $dz = 10$ m using the objective analysis method of Barnes (1994), with smoothing length scales of 3 km horizontally and 10 m vertically. Meridional velocity is then derived from gridded temperature and salinity fields via the thermal wind relation and referenced to the gridded DAC values. When we resample the mooring data at EB1 to the time steps of the glider data $\pm 1$ days, glider and mooring data agree well. Between 50 m and 1000 m the mean difference

between mooring and glider temperature and salinity is -0.029°C and $0.008\,\mathrm{g\,kg^{-1}}$, respectively. We excluded the upper 50 m as they are not resolved by the mooring data.

Although this method mitigates spatial scatter in glider data, the resulting hydrographic and velocity sections remain temporally sparse and irregular. The median interval between glider transects is approximately 3 days, with gaps ranging from a minimum of 1 day to a maximum of 166 days between missions. Performing continuous glider missions is challenging (McCarthy et al., 2020), and despite best efforts the number of transects per month varies from 0 in March to over 30 in December. As a result, the Fraser et al. (2022) product is susceptible to aliasing and is limited for studies of ESC variability.

## 2.3 Auxiliary data

To compensate for the missing velocity observations in the eastern Rockall Trough boundary region, meridional velocity fields from the Global Ocean Physics Reanalysis GLORYS12V1 (from now on referred to as GLORYS) for the period July 2014 to July 2024 were obtained from the Copernicus Marine and Environment Monitoring Service (CMEMS). GLORYS is a global ocean eddy-resolving (1/12° horizontal resolution, 50 vertical levels) reanalysis covering the altimetry period (1993 onward). Houpert et al. (2020) found that the GLORYS ocean reanalysis accurately captured the variability in the 8-month time series of meridional velocity measured by the RTADCP during its first deployment. However, the reanalysis consistently underestimated the flow strength by approximately $+7.6\,\mathrm{cm\,s^{-1}}$. To correct for this bias, they interpolated the GLORYS meridional velocities to the RTADCP location (57.1°N, 9.3°W, upper 750 m) and applied a uniform offset of $+7.6\,\mathrm{cm\,s^{-1}}$. In this study, we used both the original and bias-corrected GLORYS meridional velocity output at the RTADCP location. Our approach aims to keep the reliance on model output as minimal as possible, using it only where observational gaps cannot be filled otherwise.

Full-depth velocity and hydrographic measurements were obtained from lowered ADCP/CTD casts during Extended Ellet Line hydrographic sections between 1996 and 2017 (Houpert et al., 2020). Between 1996 and 2004, 150 kHz broadband ADCPs were used, and data were processed with software developed by Eric Firing (University of Hawaii). From 2005 onward, 300 kHz broadband ADCPs were deployed, and data were processed using the Lamont-Doherty Earth Observatory software package (Thurnherr, 2014). Absolute velocities derived from both processing methods have an estimated uncertainty of $0.02-0.03\,\mathrm{m\,s^{-1}}$ and calibrated pressure, temperature and salinity data have an accuracy of 1 dbar, 0.001°C, and 0.001, respectively (Holliday et al., 2009; Thurnherr, 2014). To remove tidal signals in the velocity data, barotropic tides at the time of each cast were subtracted using predictions from the Oregon State University Tidal Inversion Software (Egbert and Erofeeva, 2002, https://www.tpxo.net/).

To investigate oceanic conditions during high and low volume transport trough the Rockall Trough, daily geostrophic velocities and Sea Surface Height (SSH) between 2014 and 2024 are obtained from the $0.25° \times 0.25°$ CMEMS product: SEALEVEL_GLO_PH_CLIMATE_L4_MY_008_057. We derive Eddy Kinetic Energy (EKE) from the geostrophic velocities as $\frac{1}{2}(u'^2 + v'^2)$, where the prime denotes anomaly with respect to the 2014-2024 mean velocities. Illustrative mean current speeds were calculated using daily geostrophic velocities between 1993 and 2023.

Wind stress curl is derived from monthly averaged ERA5 wind stress for the period 2014–2024 to analyse atmospheric conditions associated with high and low Rockall Trough transport. The data is available on a regular lat-lon grid of 0.25° from the Copernicus Climate Change Service (C3S) Climate Data Store (CDS; H. et al., 2023).

To define the bathymetry along the Rockall Trough section, 30-arcsecond gridded data from the General Bathymetric Chart of the Oceans (GEBCO, version 20141103) were used.

## 2.4 Ellett Array Section Reconstruction

To calculate the volume, heat and freshwater transports through the Rockall Trough along the Ellett Array we first reconstruct velocity fields for three different sections: the western wedge, the mid basin and the eastern wedge (Figure 1). We derive the western wedge and mid basin section as in previous studies (Houpert et al., 2020; Fraser et al., 2022) using moored observations only. Those two sections cover the NAC branch flowing through the Rockall Trough. The eastern wedge covers the ESC and we present a new methodology which utilises both mooring and glider observations. For comparison, we also reconstruct the eastern wedge velocity field using the previous methodology (Houpert et al., 2020; Fraser et al., 2022).

### 2.4.1 Western wedge and mid basin velocity

For the reconstruction of the western wedge and mid basin transport through the Rockall Trough, we follow the methodologies of Houpert et al. (2020) and Fraser et al. (2022).

The velocity field for the western wedge extends from 12.5°W to 13.0°W along 57.5°N and is reconstructed using the gridded meridional velocity from the concatenated mooring WB1/2, which combines WB1 velocity observations above 1600 m and WB2 observations below (see Section 2.1). To represent the horizontally uniform flow observed in shipboard velocity data (Figure 1), values above 1600 m are replicated westward from WB1/2 to WB1. Below 1600 m, velocities are tapered linearly from WB1/2 to zero at the seabed. West of WB1, velocities between 250 m and 1600 m are tapered linearly to zero at the seabed, while velocities above 250 m are tapered linearly to reach zero at 12.9°W to omit the northward jet over Rockall Bank (Figure 1). The reconstructed mean western wedge velocities are shown in the supplementary information (Figure S2).

The mid basin transport per unit depth at each depth level is derived from the difference in the dynamic heights, referenced to the deepest shared level of 1760 m, between EB1 (57.1°N, 9.6°W) and WB1/2 (57.5°N, 12.5°W). This methods was establish in previous studies (Houpert et al., 2020; Fraser et al., 2022) and yields a basin-wide transport below 1,250 m of approximately -0.3 Sv, aligning with prior findings that deep northward flow is blocked by topography (Holliday et al., 2000), allowing only a small net southward transport of dense Wyville Thomson Overflow Water (-0.3 Sv; Johnson et al., 2017).

### 2.4.2 Eastern wedge velocity - previous methodology

To enable comparison, we reconstruct the eastern wedge velocity section using the method from earlier studies (Fraser et al., 2022; Houpert et al., 2020). Since the new methodology retains some components of this approach, we briefly outline the original method first.

The eastern wedge subsection extends from mooring EB1 at 9.6°W to 9.2°W along 57.1°N (Figure 1b). As mentioned above, GLORYS output is used at the RTADCP location to compensate for missing velocity observations of the ESC core, while keeping reliance on model output to an absolute minimum (Houpert et al., 2020; Fraser et al., 2022). Because Houpert et al. (2020) found that the reanalysis product consistently underestimated observed meridional velocities by approximately +7.6 cm s$^{-1}$, this uniform offset is applied to the GLORYS output interpolated at the RTADCP location (57.1°N, 9.3°W, upper 750 m).

In the old method, the bias-corrected GLORYS and gridded EB1 meridional velocities were linearly interpolated between EB1 and the RTADCP location for depths shallower than 750 m. East of RTADCP, the GLORYS velocities were tapered linearly to zero at 9.2°W, marking the eastern end of the section. For depths below 750 m, the gridded EB1 velocities were extended horizontally eastward to the seabed.

### 2.4.3  Eastern wedge velocity from glider and mooring observations

Gliders capture the Rockall Trough eastern wedge velocity field with high spatial resolution, while moored EB1 current meters offer excellent temporal resolution and longer-term coverage (from 2014 onward in this instance). This motivates a combined approach that leverages the strengths of both datasets. We are guided by the methodology of Brandt et al. (2014, 2016, 2021), who integrate long-term mooring observations with high-resolution ship sections to construct a comprehensive velocity product. We describe the methodology in detail below highlighting any difference to the original methodology by Brandt et al. (2014, 2016, 2021).

We first linearly interpolate the glider transect onto the 20-dbar vertical grid of the moored data and apply an empirical orthogonal function (EOF) analysis to identify the dominant modes of variability in the eastern wedge meridional velocity field, based on 166 glider transects (Figure 3). The temporal mean is removed from each glider section prior to performing the EOF analysis and the results are discussed in Section 3.1.1.

Our approach slightly differs from Brandt et al. (2014, 2016, 2021) who applied the Hilbert empirical orthogonal function (HEOF) method to repeated ship sections to better capture the vertical displacement and spatial migration of current cores. In the HEOF approach, a Hilbert transformation is applied to the input field before performing the EOF analysis, allowing spatial propagation features—such as meridional shifts of a current—to be represented within a single mode. In our study, we evaluated both EOF and HEOF approaches with similar results and chose to use EOF patterns for simplicity.

The second step in reconstructing the eastern wedge velocity field is to regress the EOF patterns ($X(x,z)$) onto meridional velocity anomalies from the EB1 mooring and GLORYS output at the RTADCP location with respect the temporal 2014–2024 mean. We are guided by the previous methodology in using GLORYS exclusively at this location to minimize reliance on model output and because of the validation against observations at that location (Houpert et al., 2020). In the new approach, bias correction of GLORYS is not applied. Through the regression, we estimate a anomalous velocity section $v'(x,z,t)$ for each time step, such that

$$v'(x,z,t) = X(x,z) \cdot \alpha(t) \tag{1}$$

We must therefore obtain the regression coefficient $\alpha$. We combine the meridional velocity anomalies from EB1 and GLO-
RYS at each time step to get $v'_{loc}(t)$, select values of the EOF pattern at the location of EB1 and RTADCP to get $X_{loc}$ and find
$\alpha(t)$ using a least squares solution:

$$\alpha(t) = (X_{loc}^T X_{loc})^{-1} X_{loc}^T v'_{loc}(t). \tag{2}$$

The full velocity field is finally derived by summing the reconstructed anomaly fields $v'(x,z,t)$ from the selected EOF modes
and adding the temporal mean of the glider section for each time step. Choosing the number of EOF modes requires balancing
accuracy and complexity, as EOF patterns are statistical and may not fully represent physical processes. For our transport
estimates, we evaluated the EOF modes in detail (Section 3.1.1) and found that the first two EOFs capture the variability of the
velocity field for our transport estimates best. This approach defines the mean velocity field above 1000 m as the multi-year
glider mean, making bias correction of GLORYS redundant.

In the final step, the velocity field below 1000 m is constructed by extending gridded EB1 velocities horizontally eastward
to the seabed, following Houpert et al. (2020) and Fraser et al. (2022).

## 2.5 Volume, heat and freshwater transport

We calculate the volume, heat and freshwater transports for each section as described below. Total transports through the
Rockall Trough are then calculated as the sum of the transports in the western wedge, the mid basin and the eastern wedge.

The volume transport $Q$ is calculated by spatially integrating a velocity field:

$$Q = \int_{-H}^{0} \int_{x_1}^{x_2} v(x,z) \, dx \, dz \tag{3}$$

where $v$ is the velocity component perpendicular to the section, $x$ and $z$ are the along-section and depth coordinates, $x_1$ and $x_2$
are the section endpoints and $H$ is water depth.

The heat transport $Q_h$ is given by

$$Q_h = \rho_{ref} C_p \int_{-H}^{0} \int_{x_1}^{x_2} v(x,z) (\Theta(z) - \Theta_{ref}) \, dx \, dz \tag{4}$$

where $\rho_{ref} = 1027.5 \, \text{kg}^{-1} \, \text{m}^{-3}$ is the reference density, $C_p = 3991 \, \text{J} \, \text{kg}^{-1} \, \text{C}^{-1}$ is the specific heat capacity, and $\Theta_{ref} = 7.137 \, ^\circ\text{C}$ is the reference conservative temperature, defined as the depth-averaged temporal mean profile at WB1/2 over the
full observation period. $\Theta(z)$ denotes the conservative temperature profile, taken from WB1/2 for the western wedge, from the
average of WB1/2 and EB1 for the mid basin, and from EB1 for the eastern wedge.

The freshwater transport $Q_f$ is calculated as follows:

$$Q_f = -\int_{-H}^{0} \int_{x_1}^{x_2} \left( v(x,z) \frac{S(z) - S_{ref}}{S_{ref}} \right) dx \, dz \tag{5}$$

where $S_{ref} = 35.342 \, \mathrm{g \, kg^{-1}}$ is the reference absolute salinity, defined as the depth-averaged temporal mean profile at WB1/2 over the full observation period. $S(z)$ denotes the absolute salinity profile, taken from WB1/2 for the western wedge, from the average of WB1/2 and EB1 for the mid basin, and from EB1 for the eastern wedge.

The two moorings (WB1 and WB2) used to create the WB1/2 temperature and salinity profiles are located in the southward-flowing current west of the NAC (Figure 1b). By referencing temperature and salinity to the depth-averaged temporal mean of these profiles over the full mooring period, we effectively set the mean heat and freshwater transport through the western wedge near zero. Please note, because the triangular-like geometry of the western wedge weights the saltier and warmer surface layers more heavily than the reference profile, a small residual but neglectable mean transport remains. Consequently, the transports calculated for the mid-basin and eastern wedge primarily reflect signals from the northward flowing NAC and ESC, respectively.

## 2.6 Accuracy of volume transport calculations - previous methodology

Houpert et al. (2020) conducted a comprehensive error analysis of the transport estimates using a Monte Carlo approach to assess the impact of instrument errors, and evaluated methodological uncertainties by subsampling data from lowered ADCP and CTD data section from the Extended Ellet Line cruises, alongside climatology data from MIMOC (Monthly Isopycnal and Mixed-layer Ocean Climatology; Schmidtko et al., 2013) along the section. Houpert et al. (2020) define a mean bias error as the mean difference between full and subsampled transport estimates and a root-mean-square error (RMSE) as the standard deviation of these differences across sections.

For the western wedge transport, Houpert et al. (2020) reported a mean bias error of -0.30 Sv and a RMSE of 0.63 Sv, primarily due to horizontal extrapolation of current meter data and measurement accuracy. Errors in the mid basin transport were attributed to vertical gridding of CTD data, surface extrapolation, and measurement accuracy. For this region, the mean bias error was 0.11 Sv and the RMSE of 0.34 Sv. Instrument failures and losses introduced additional variability in the mid basin: bias errors were -0.30 Sv (2014–2015) and 0.41-0.47 Sv (2016–2017), with RMSE values of 0.68 Sv (2014–2015) and 0.34-0.36 Sv (2016–2017). For the eastern wedge, the mean bias error was 0.21 Sv and the RMSE was 0.59 Sv, reflecting uncertainties from horizontal extrapolation of current meter data and reliance on GLORYS output at the RTADCP location.

Houpert et al. (2020) estimated errors for the total Rockall Trough transport time series by combining uncertainties from the western wedge, mid basin, and eastern wedge. For periods of optimal data return (2015–2016, since mid-2017), they found a mean bias error of 0.03 Sv and an RMSE of 0.93 Sv. However, data loss due to instrument failure increased uncertainty in the mid basin transport, affecting the total transport uncertainties with mean bias of -0.39 Sv (0.38 Sv) and RMSEs of 1.10 Sv

(0.94 Sv) for the earlier (later) period of instrument loss. For further details, we refer readers to section 4 in Houpert et al. (2020).

## 2.7 Accuracy of heat and freshwater transport calculations

To evaluate whether hydrographic measurements at mooring locations are sufficient for calculating heat (Equation 4) and freshwater transport (Equation 5) across the Rockall Trough sections, we sub-sampled ship-based and glider-based hydrographic sections at the mooring positions and compared the resulting transports to those derived from the full sections (Table S1). For the western wedge and mid basin, we used ship sections; for the eastern wedge, we used glider sections. Only ship sections that included all Extended Ellet Line stations (Figure 1) for the respective regions were considered. Prior to transport calculations, ship sections were linearly interpolated onto a regular 5 km grid. Table S1 in the supplementary information summarises the mean and standard deviation for both approaches, along with the mean bias error and RMSE between full-section and profile-based estimates. Gernerally the mean bias error of the western and eastern wedge are small ($1\% - 6\%$ of transport mean) and also the RMSE error are clearly smaller than one standard deviation of the transports ($4\% - 8\%$) with a moderate RMSE for the freshwater transport in the eastern wedge ($19\%$ of one standard deviation). The mean bias error ($22\% - 31\%$ of mean) and RMSE ($25\%$ of one standard deviation) for the mid basin heat and freshwater transport are higher, but given the large natural variability we find the results acceptable.

## 3 Results

This section is divided into two parts. First, we focus on the eastern wedge and the ESC: validating the new methodology for reconstructing its velocity field, examining its mean strength and variability in glider and moored observations, and assessing the continuous transport reconstruction for the eastern wedge section. Second, we analyse the 10-year records of volume, heat, and freshwater transport for the full Ellett Array section in the Rockall Trough, exploring trends, variability from intraseasonal to interannual scales, and associated oceanic and atmospheric conditions. The discussion section follows the same structure to facilitate cross-referencing.

### 3.1 Part I – Eastern Wedge (ESC) reconstruction

Extending the Ellett Array dataset to 2024 provides, for the first time, a temporal overlap between glider and moored observations of the ESC long enough to allow a direct comparison of transport estimates in the upper 1000 m from glider-derived velocities and from mooring-based reconstructions using both the previous and current methods.

#### 3.1.1 Comparison and validation of reconstruction approaches

The mean glider section shows that the eastern wedge velocity field is dominated by the northward flowing ESC, with a southward flowing undercurrent below 800 m (Figure 3a). To reconstruct the eastern wedge velocity field with the new methodology we first performed an EOF analysis of the glider sections (Figure 3b-d). The first EOF explains 54% of the total variance and

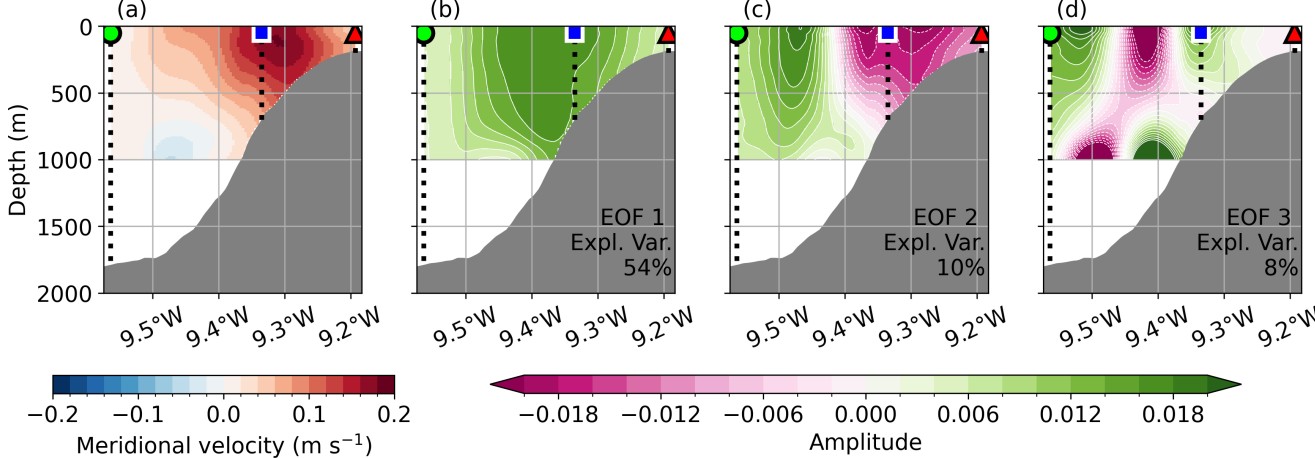

**Figure 3.** (a) Mean meridional velocity from 166 glider transects along the eastern wedge (2020–2023). (b) First three Empirical Orthogonal Function (EOF) patterns and their explain variance derived from the glider-based meridional velocity sections. Mooring EB1 is marked by a black dotted line with a green circle; the RTADCP (2014–2015) is marked by a black dotted line with blue rectangle; the section endpoint of the eastern wedge is marked by a black dotted line with red triangle. Please note that there are no glider data below 1000 m.

represents a coherent strengthening or weakening of the entire section, with peak amplitudes between 9.3°W and 9.5°W just west of the mean ESC core. Near-zero values appear below 800 m between 9.4°W and 9.5°W, corresponding to the core of a southward-flowing undercurrent (Fraser et al., 2022). The second EOF explains 10% of the variance and shows a dipole pattern centered at 9.4°W. This pattern resembles a typical cross section of an eddy. Again near-zero values can be found at the location of the southward flowing undercurrent core. Higher-order EOF modes (mode 3 and above) explain less than 8% of the variance individually and display more complex, less interpretable patterns. Overall, the EOF results indicate that the dominant variability in the ESC is primarily due to temporal strengthening and weakening slightly offset to its mean position.

To validate the new methodology and determine how many EOF patterns are needed to capture ECS variability, we compared velocity sections and transports from glider observations with those reconstructed using the old and new eastern wedge methodologies both of which use EB1 observations and GLORYS output. The mean meridional velocity section from the two different eastern wedge reconstructions are shown in Figure 4. The spatial structure of the upper 1000 m in the old methodology closely resembles EOF mode 1 (Figure 3b), whereas–by definition–the temporal mean of the upper 1000 m in the new methodology resemble the mean glider section. This constraint ensures that the new approach reflects the observed mean state of the ESC and undercurrent, which was not guaranteed in the previous method and likely makes the spatial structure more realistic.

To enable direct comparison, we resampled the reconstructed transport estimates by averaging data within $\pm 1$ day of each glider time step. For the old method, correlation of reconstructed transports with glider-derived transports is $R = 0.75$ with an RMSE of 1.25 Sv (Figure 5a). The new method achieves similar correlation when using the first two EOF modes (Fig-

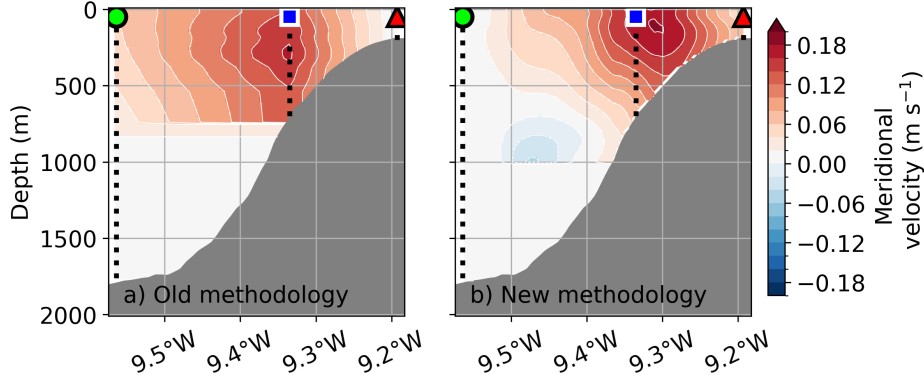

**Figure 4.** Averaged meridional velocity (2014-2024) for the eastern wedge using a) old methodology from Houpert et al. (2020) and Fraser et al. (2022) and b) the new methodology including glider observations. Black dotted line with green circle marks mooring EB1, black dotted line with blue square marks the position of the RTADCP, black dotted line with red triangle marks the eastern limit of the section. Please note that values below 1000 m are near zero and therefore barely visible.

ure 5b), with slightly improved RMSE (1.21 Sv) and standard error (0.05 Sv). To test reconstruction without GLORYS output,

we repeated the same validation by reconstructing the velocity field using only EB1 observations. This results in a weak correlation with observed glider transports (Figure 5c). Correlation sections of sub-sampled glider velocity profiles at EB1 and RTADCP locations with velocities across the full glider section (Figure S2) highlight the importance of both locations, as correlations drop rapidly within $\sim 0.1°$ longitude. Based on these tests, we use two EOF modes in the final reconstruction because they provide the best balance between capturing variability and maintaining strong correlation when combined with the EB1

observations and GLORYS output. Physically, this choice is supported by the interpretation of EOF mode 1 as representing large-scale strengthening or weakening of the slope current (Huthnance, 1984; Huthnance et al., 2022; Marsh et al., 2017) which is also represented by the old methodology, while EOF mode 2 reflects modifications by the region's high mesoscale activity (Gary et al., 2018).

Overall, in the new approach produces a mean velocity structure that captures a realistic ESC and an undercurrent below

800 m—features absent in the previous methodology (Figure 4).The best results are achieved by combining the first two EOF patterns of glider sections with observed velocities at EB1 and GLORYS output at the RTADCP to reconstruct the upper 1000 m of the eastern wedge. The new approach still relies on GLORYS output, as reconstructions based solely on mooring and glider observations proved unrealistic. An improvement of the new methodology, however, is that it eliminates the need for bias correction of GLORYS output, which in the old approach was based on only eight months of ADCP data.

### 3.1.2 ESC volume transport time series

This section focuses on the ESC volume transport time series derived from glider observations and the two mooring-based reconstructions. All three estimates show good agreement (Figure 6a), with both reconstructions correlating well with glider-

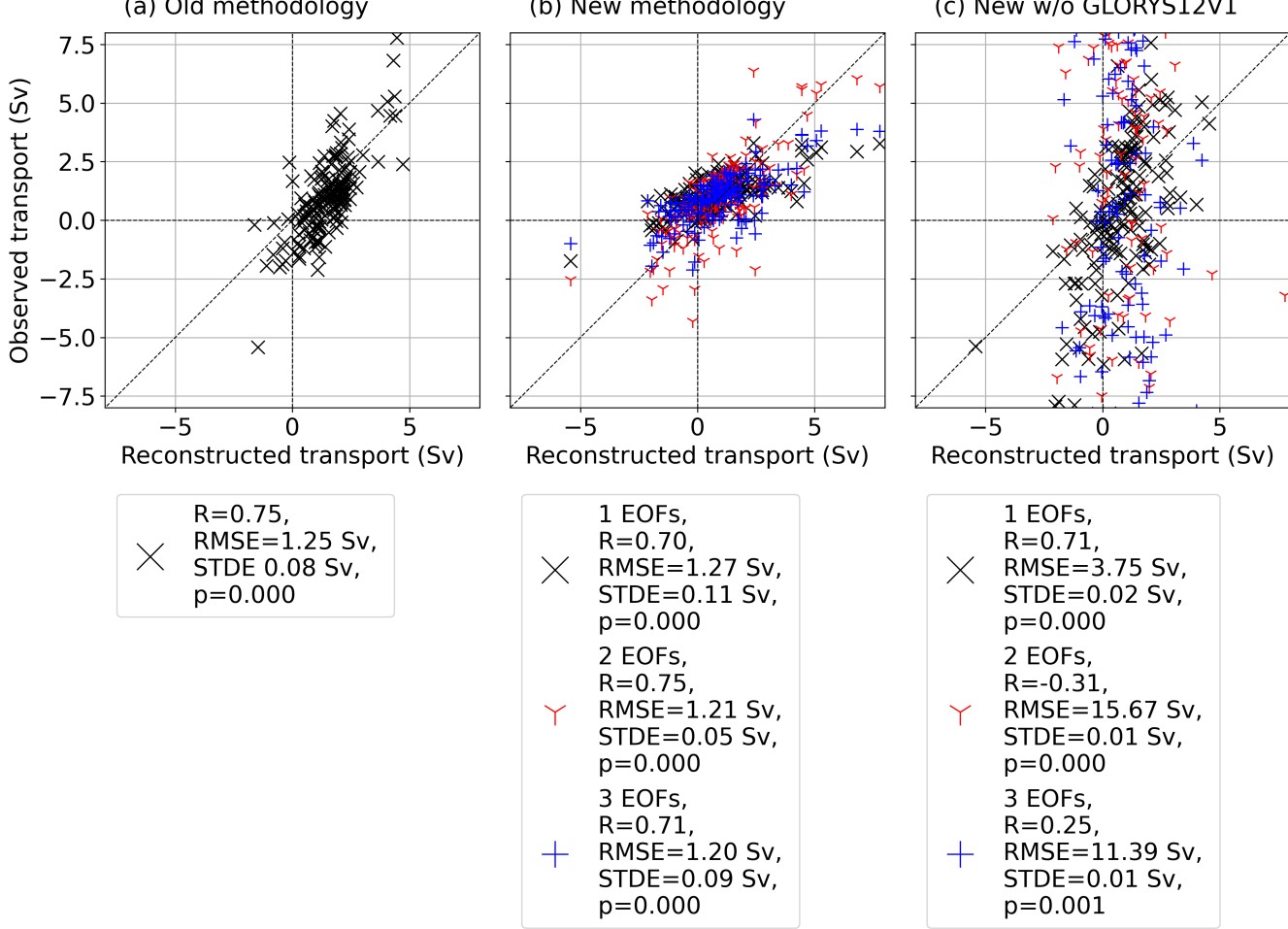

**Figure 5.** Linear regression of mooring-based reconstructions versus glider-derived transports. Panels show (a) the old methodology, (b) the new methodology using EB1 observations and GLORYS output at RTADCP, and (c) the new methodology using only EB1 observations. Statistics shown: R = Pearson correlation coefficient; RMSE = Root Mean Square Error; STDE = Standard Error; p = two-sided p-value for the null hypothesis that the slope equals zero.

derived transport (both $R = 0.75$, Figure 5). Between April 2020 to February 2023, the mean transport for the glider and the new eastern wedge reconstruction is 1.0±0.3 Sv (mean±1 standard error), while the old methodology gives 1.5±0.2 Sv. Although the agreement is marginal, the estimates agree within their respective uncertainties. Notably, extreme transport events observed by the gliders—such as in December 2020 and May 2021—are suppressed in the old reconstruction and better captured by the new approach.

The variability of the new reconstruction depends, as in the old methodology, mainly on moored observations at EB1 and GLORYS output at the RTADCP location, but is tuned by glider-derived transport variability. By regressing moored and GLORYS velocities onto the two dominant EOF modes of the glider data, variability associated with these patterns is emphasised. This is evident in the power density spectra comparing the old and new reconstruction approaches (Figure 7), where higher-frequency variability with periods between 23 and 120 days is elevated in the new reconstruction, likely reflecting the better spatial representation of mesoscale activity introduced through EOF mode 2. The importance of EOF mode 2 is further supported by the fact that using only EOF mode 1 in the new methodology fails to reproduce these extreme events (not shown).

The seasonal cycle of the ESC appears weak, as the standard deviation for each month is nearly as large as the seasonal signal itself (Figure 6d). Sparse and irregular glider sampling—especially gaps in March—limits seasonal resolution. Seasonal cycles (Figure 6c) show a May peak in the glider data, absent from reconstructions over the full overlap period. However, this peak appears when the new reconstructed transport is resampled to glider time steps (Figure 6b-c), driven by the strong positive transports in May 2021. Note that, although weaker, this peak in the seasonal cycle also appears in the reconstructed transport derived with the old approach when resampled onto the glider time steps (not shown). Over the full mooring period (2014–2024), the weak seasonal cycle shows a January minimum, a secondary August minimum, and enhanced transports in spring to early summer, and autumn.

### 3.2 Part II – Full Rockall Trough transports 2014-2024

### 3.3 Rockall Trough volume, heat and freshwater transports 2014-2024

The northward volume transport through the Rockall Trough has a mean of 4.7±0.5 Sv between 2014 and 2024 (Figure 8a), which is within the uncertainty of previous transport calculations for shorter periods (Houpert et al., 2020; Fraser et al., 2022). As in these previous studies, the mid basin dominates the volume transport with a mean of 5.0±0.5 Sv, while the western and eastern wedges have mean volume transports of -1.5±0.3 Sv and 1.3±0.2 Sv, respectively. The total northward heat and freshwater transport is (4.5±0.3)·$10^{-2}$ PW and (-1.9±0.2)·$10^{-2}$ Sv, respectively. For the mean heat and freshwater transports, the mid basin accounts for about 75% and the eastern wedge accounts for about 25% while—as defined by our choice of reference values—the western wedge contribution is near zero (Figure 8b-c).

The 10-year Rockall Trough time series of volume, heat, and freshwater transport displays variability on the order of several Sverdrups on intraseasonal to interannual time scales (Figure 9) consistent with previous studies (Houpert et al., 2020; Fraser et al., 2022). This large variability makes detection of linear trends challenging and for the total transport time series we do not find any significant trend (calculated following Hamed and Rao, 1998). While opposing trends at the 5–10% confidence

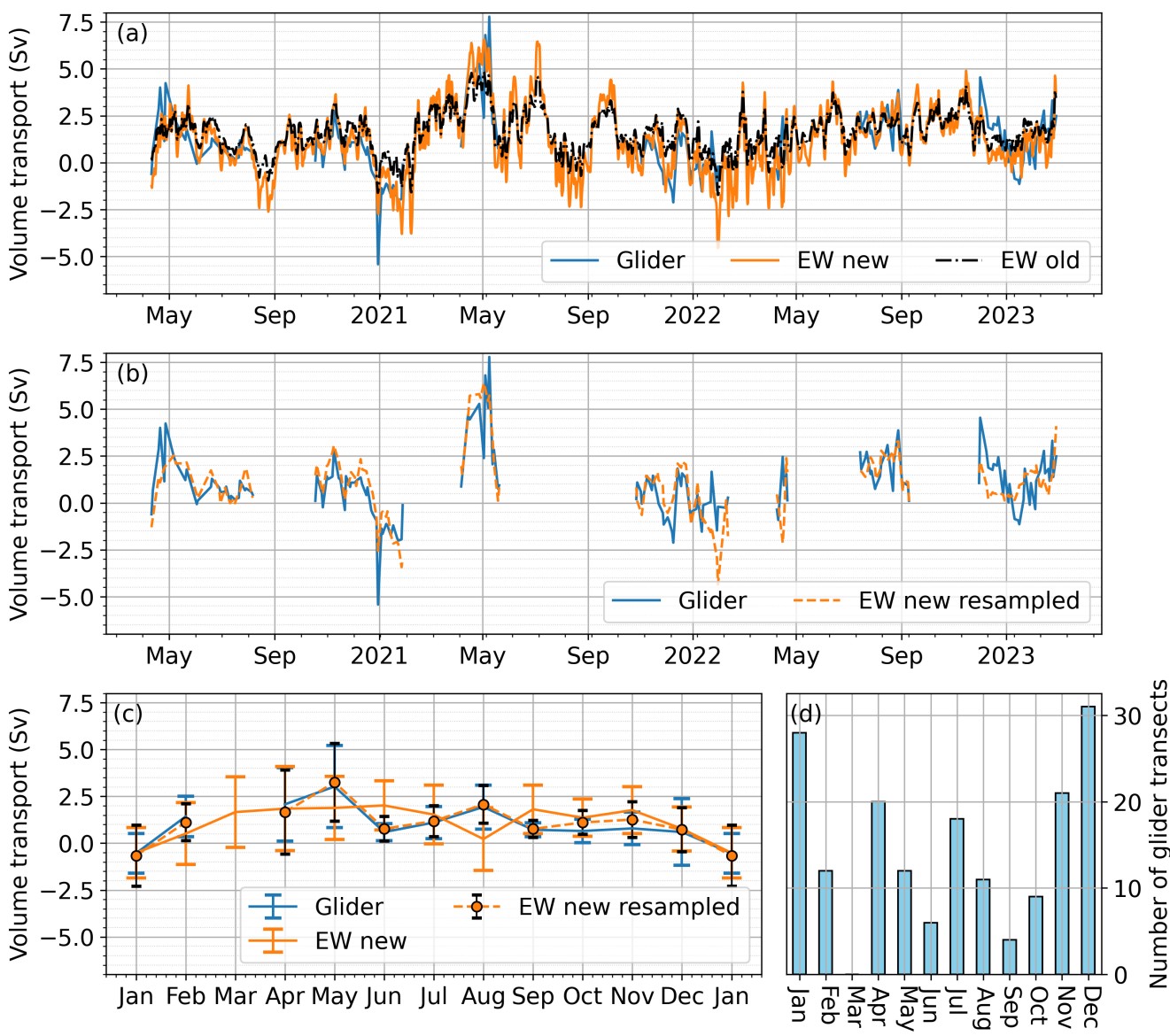

**Figure 6.** (a) Time series of volume transport for the upper 1000 m derived from meridional velocity section based on glider transects (blue line), the new eastern wedge reconstruction (EW new, orange line), the old eastern wedge reconstruction (EW old, black dotted line). (b) Transport time series of the new approach averaged ±1 day on the time step of the glider observations (EW new resampled, orange dashed line). Glider transport are shown in blue again for reference. (c) Monthly mean seasonal cycle for glider (blue line), the new eastern wedge reconstruction (EW new, orange line) and the resampled new eastern wedge reconstruction (EW new resampled, orange dashed line). The error bars mark ±1 standard deviation. (d) Number of glider transects per month of the year.

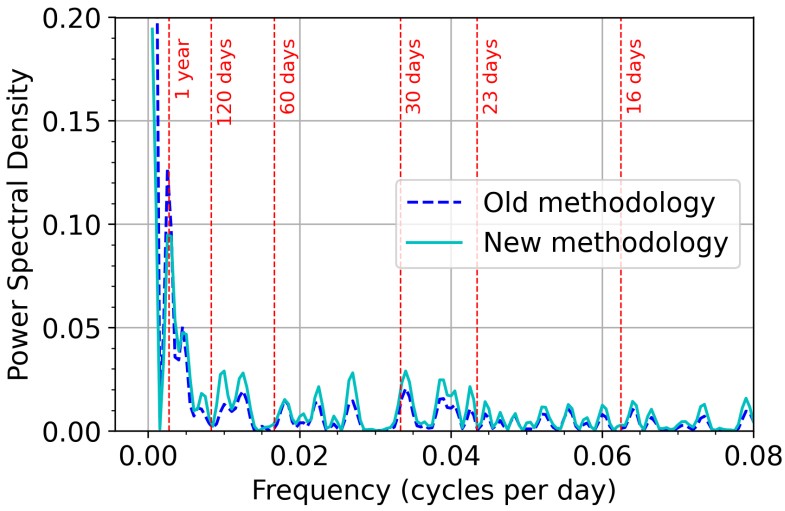

**Figure 7.** Power density spectra of eastern wedge transports in the Rockall Trough reconstructed using the old (dashed blue line) and the new methodology (solid cyan line) for the period 2014–2024. Vertical dashed red lines mark selected periods.

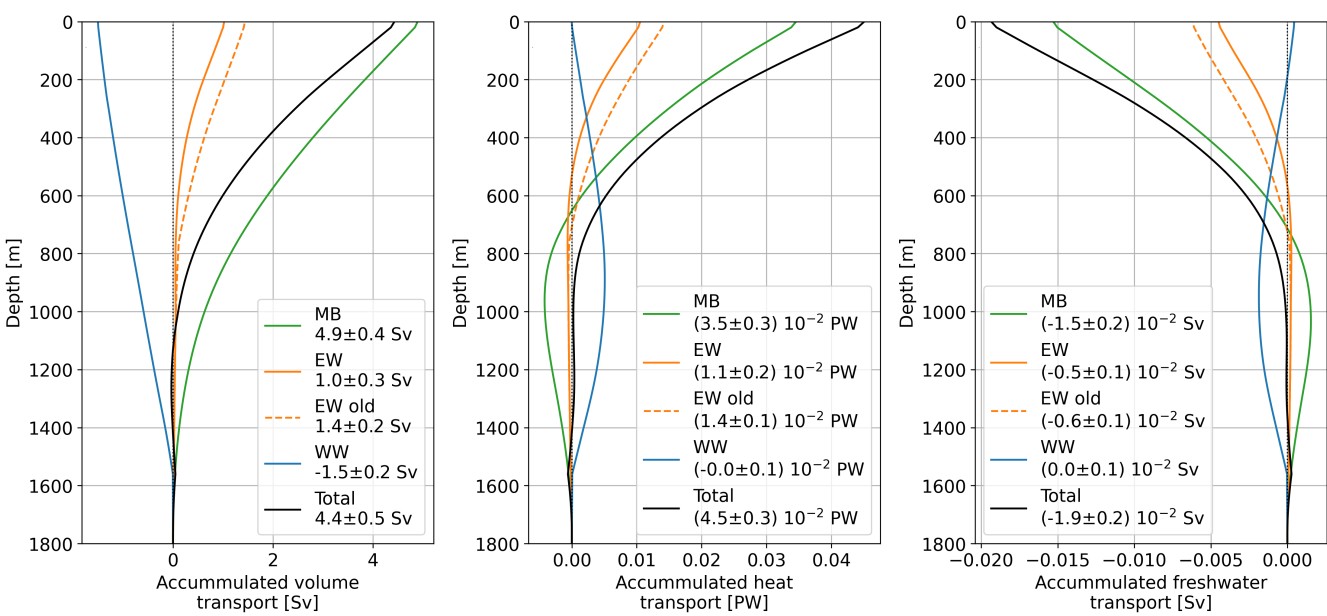

**Figure 8.** Depth-accumulated mean a) volume, b) heat and c) freshwater transports in the RT. Values are shown for the mid basin (MB) in green, the eastern wedge (EW) in orange (solid for the new, dashed for the old methodology), the western wedge (WW) in blue and the total transport in black.

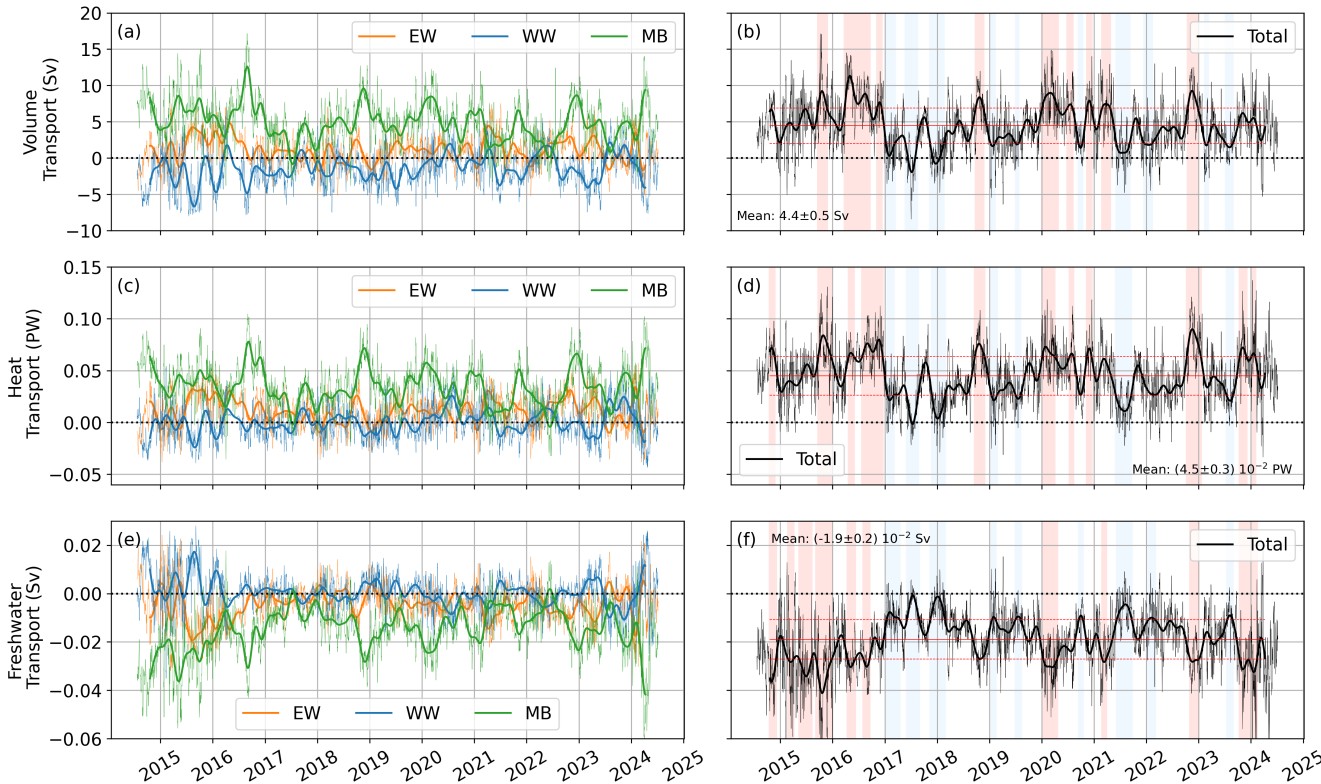

**Figure 9.** Rockall Trough (a,b) volume, (c,d) heat, and (e,f) freshwater transports. Panels (a,c,e) show transports for the eastern wedge (EW, orange), western wedge (WW, blue), and mid basin (MB, green); panels (b,d,f) show the total (black). Thin lines indicate unfiltered data, bold lines the 90-day low-pass-filtered data. In (b,d,f), red solid lines denote the mean total transport, and red dashed lines the $\pm 1$ standard deviation. In (b,d), red shading marks periods when the filtered time series exceed the mean $+1$ standard deviation, and blue shading when they fall below the mean $-1$ standard deviation. In (f), the shading is reversed.

level occur in the western wedge and mid-basin transports (Table S2), these are small compared to the pronounced interannual variability and may reflect the influence of multi-year anomalies rather than long-term change. Notably, the moored temperature and salinity time series capture the mid-2010s subpolar cold freshwater anomaly (Holliday et al., 2020; Fox et al., 2022), which reached the northern Rockall Trough in 2017 (Figure 2 and S4; Fraser et al., 2022), followed by a recovery in salinity during the most recent two years associated with warm temperature anomalies. In the following, we focus on interannual variability in volume transport and then examine variability in heat and freshwater transports.

The NAC dominates variability in total Rockall Trough volume transport with the mid-basin transport explaining 40% of the variance (R=0.63) in the detrended 90-day low-pass filtered total transport time series. In the following all correlations refer to the detrended 90-day low-pass filtered time series. The ESC is the secondary driver explaining 20% of the variance (R=0.45) while variability through the western wedge is negligible (R=0.18). The influence of ESC variability is evident in the seasonality

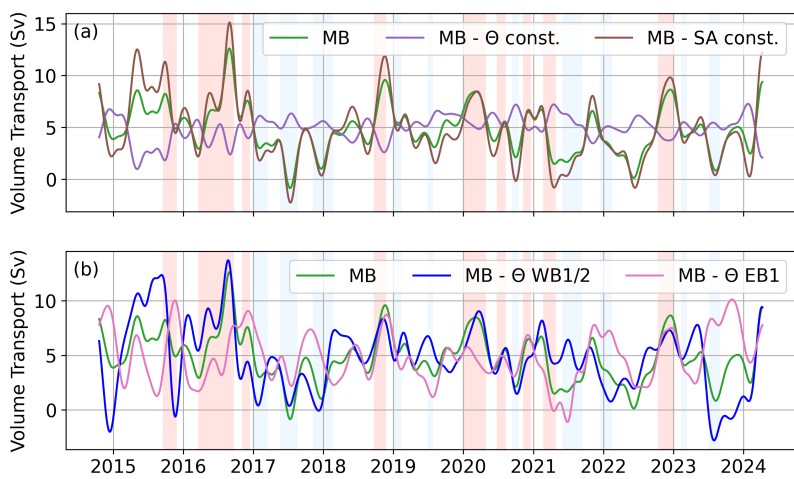

**Figure 10.** Time series of 90-day low-pass filtered volume transport through mid basin (MB) (a) isolating changes in salinity (Θ const., purple line) or temperature (SA const., brown line) and (b) isolating changes in temperature at WB1/2 (blue line) or EB1 (light pink line). The green line shows the mid basin volume transport as shown in Figure 9 for reference. Red (blue) shading mark period when the total Rockall Trough volume transport is higher (lower) than the temporal mean+1 standard deviation (−1 standard deviation).

of extreme transport events (Figure S5) which for example notably increase the occurrence of low total transport events during December to February. We find a significant anti-correlation of $R = -0.44$ between mid-basin and western wedge volume transports. The eastern wedge transport is not correlated with the mid-basin (R=-0.02) and only weakly anti-correlated with the western wedge (R=-0.18) indicating that the ESC is dynamically distinct from the NAC branch in the Rockall Trough.

To examine how temperature and salinity changes influence transport, we isolate their contributions from the eastern and western boundaries to the mid-basin transport which dominates total transport variability and exhibits the strongest trend within the 10-year period (Figure 10). As shown in Fraser et al. (2022), temperature changes primarily drive mid-basin interannual variability. Isolating temperature changes at either boundary for mid-basin transport (Figure 10b) does not reveal a clear dominance of one side during strong or weak transport events. While no significant trend is found in the salinity-driven transport 415 time series, we detect a significant decrease of $-3.39\,\mathrm{Sv}$ in the temperature-driven transport time series over the full 10-year observation period.

To assess whether temperature-driven changes represent a long-term trend or multi-year anomalies we split the time series at the beginning of 2022 when positive subsurface temperature and salinity anomalies developed at the mooring sides (Figure S4). Trends are now expressed in Sv per year rather than Sv per decade. Between 2014–2021 mid-basin transport decreased 420 significantly (5% confidence level) by $-0.29\,\mathrm{Sv/year}$ dominated by temperature-driven changes of $-0.51\,\mathrm{Sv/year}$ and partly offset by a salinity-driven strengthening of $0.21\,\mathrm{Sv/year}$. After 2022 all three time series indicate a strengthening trend but only the total mid-basin transport trend is significant at the 10% confidence level with $0.85\,\mathrm{Sv/year}$. These results suggest that apparent long-term trends in transport are linked to the arrival and subsequent decay of the mid-2010s subpolar cold freshwater

anomaly in the northern Rockall Trough. While the freshwater anomalies acted to strengthen the flow between 2014 and 2021
temperature-driven transport anomalies dominated causing a reduction in mid-basin transport during that period. Since 2022
positive temperature and salinity anomalies act to strengthen the flow.

Heat and freshwater transports are generally dominated by changes in volume transport (Table 1). This is also represented
in the occurrence of extreme transport events which we define as transport higher (lower) than the temporal mean $+1$ standard
deviation ($-1$ standard deviation; Figure 9b, d, f). However, changes in the depth-integrated temperature (for heat transport)
and salinity (for freshwater transport) can contribute as secondary drivers within the Rockall Trough (Table S3). We find that
the depth-integrated averaged salinities of WB1/2 and EB1 are anti-correlated with the total and mid basin freshwater transport
(R=-0.47) and explain 22% of freshwater transport variability (Table S3). For the eastern wedge, we find a weak to moderate
anti-correlation of the depth-integrated salinities at EB1 (R=-0.33), while the freshwater transport of the western wedge is
not correlated with salinity changes in the western Rockall Trough. However, here we find a positive correlation between the
depth-integrated temperatures of WB1/2 and the heat transport (R=0.46). This indicates, while volume transport dominates,
temperature and salinity anomalies may amplify or dampen these extremes, especially in the mid basin for freshwater and
western wedge for heat transport.

**Table 1.** Pearson correlation coefficient (R), p-value, and explained variance ($R^2$) for the relationship between detrended 90-day low-pass
filtered volume ($Q$), heat ($Q_h$) and freshwater ($Q_f$) transport.

|  | $Q$ **vs** $Q_h$ | $Q$ **vs** $Q_f$ |
| --- | --- | --- |
| **Total** |  |  |
| Pearson R | 0.90 | -0.84 |
| P-value (p) | 0.00 | 0.00 |
| $R^2$ | 82% | 71% |
| **Western wedge** |  |  |
| Pearson R | 0.93 | -0.93 |
| P-value (p) | 0.00 | 0.00 |
| $R^2$ | 86% | 86% |
| **Mid basin** |  |  |
| Pearson R | 0.93 | -0.86 |
| P-value (p) | 0.00 | 0.00 |
| $R^2$ | 87% | 73% |
| **Eastern wedge** |  |  |
| Pearson R | 0.96 | -0.96 |
| P-value (p) | 0.00 | 0.00 |
| $R^2$ | 93% | 93% |

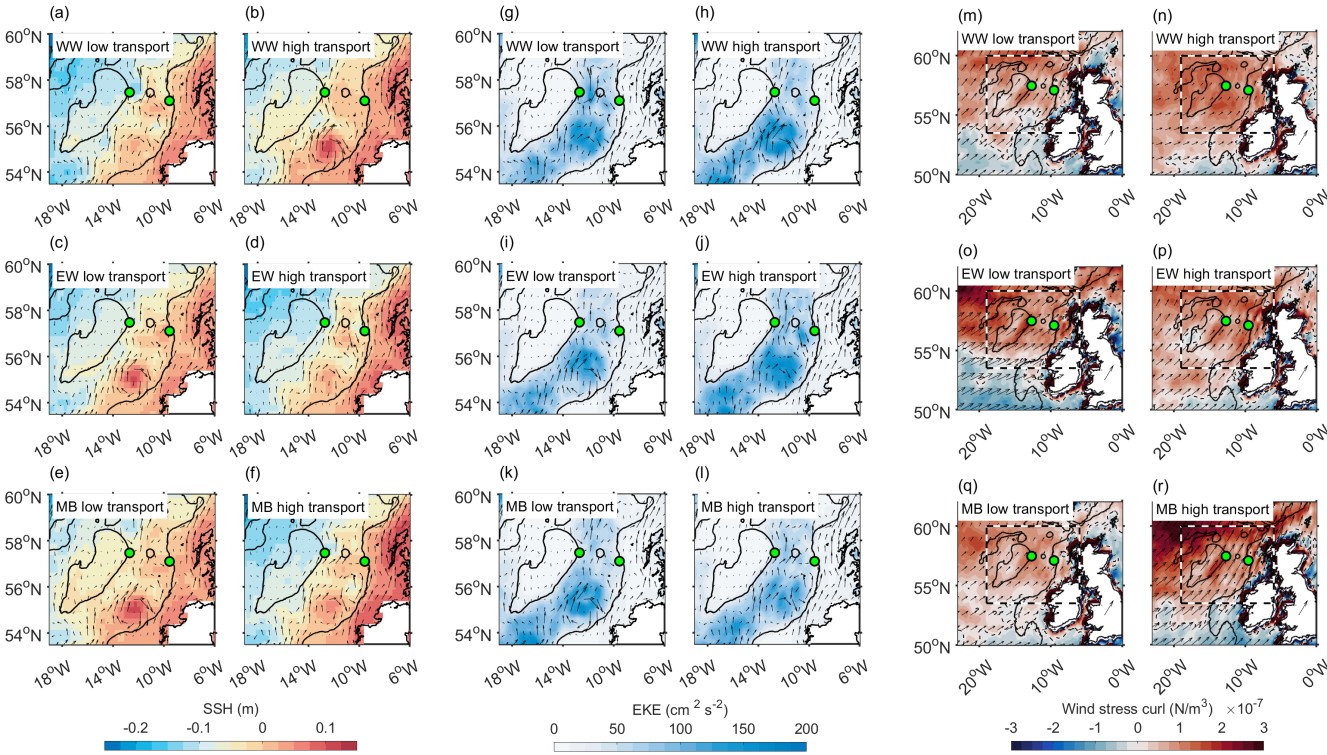

**Figure 11.** Composites for high (mean + 1 standard deviation) and low (mean - 1 standard deviation) transport events in the western wedge (WW; top panels), eastern wedge (EW; middle panels), and mid-basin (MB; bottom panels). Colour shading indicates (a–f) sea surface height (SSH), (g–l) eddy kinetic energy (EKE), and (m–r) wind stress curl. Arrows in panels (a–l) show geostrophic velocity, while arrows in panels (m–r) show horizontal wind stress (reference arrow on land corresponds to $0.1\,\mathrm{N\,m^{-2}}$ for both zonal and meridional components). Note the larger spatial domain in panels (m–r); the black-and-white dashed box marks the area shown in panels (a–l). Green circles indicate mooring positions WB1 (12.7°W) and EB1 (9.6°W).

### 3.3.1 Ocean and atmosphere state during transport extremes

We investigate the oceanic and atmospheric conditions in the Rockall Trough during high and low transport events by calculating composites of SSH, EKE, wind stress and wind stress curl (Figure 11). Because the composites for total and mid basin transport are similar, we show only results for mid basin, western and eastern wedge and mention any differences for the total transport in the text.

For total and mid basin transport, high events are characterised by elevated SSH over the shelf and reduced SSH across Rockall Bank. These conditions coincide with a weak anticyclonic eddy south of 56°N and anticyclonic circulation in the center of the Rockall Trough southwest of EB1. In contrast, low transport shows a weaker SSH gradient between the shelf and Rockall Bank associated with a stronger anticyclone south of 56°N and an eastward-shifted anticyclonic feature southwest of

EB1. We also find slightly elevated EKE west of EB1 for high total transport while there are no noticeable differences in EKE for the mid basin composites. Wind stress curl during high mid basin transport exhibits a stronger positive anomaly particularly towards the northwest of the Rockall Trough with the transition from negative to positive curl located south of 55°N. During low mid basin transport we find notably weaker wind stress and wind stress curl.

Western wedge transport is anti-correlated with mid basin transport so we compare composites of high western wedge transport to low mid basin transport and vice versa. The northern anticyclone near EB1 is more centred within the Rockall Trough during high transport and displaced eastward during low western wedge transport but the overall SSH patterns are generally similar to the respective results of the mid basin transport. For EKE we find elevated values in the west just east of moorings WB1 and WB2 during low transport. The wind stress curl transition zone lies further south during high western wedge transport similar to the low mid basin transport composite. Low transport in the western wedge appears to be associated with stronger meridional winds similar to high mid basin transport but magnitudes are weaker and do not show intensification towards the northwest.

For eastern wedge, the SSH pattern during high transport is similar to the mid basin case except for a local depression west of mooring EB1. Low transport shows reduced SSH over Rockall Bank compared to the low mid basin transport composite. We find high EKE west of EB1 during high eastern wedge transport which is not present during low transport. Wind stress curl indicate that during high transport the transition from negative to positive curl occurs at about 50°–51°N with weaker positive wind stress curl and a more southerly wind direction. Low transport is characterised by a sharp transition near 55°N strong positive curl in the Rockall Trough that intensify towards the northwest and pronounced southwesterly winds. Note that the wind direction here has a stronger westerly component than during high mid basin transport.

## 4    Discussion

### 4.1    Part I – Eastern Wedge (ESC) reconstruction

The new velocity field reconstruction of the eastern wedge shows an improved mean flow that captures a southward undercurrent below 800 m (Figure 3a and 4). This is expected, as the mean is largely determined by the mean glider section. A limitation is that the 10-year time series in the Rockall Trough is mainly based on a three-year glider mean. However, although the regression of glider EOF patterns onto the moored and GLORYS velocities primarily sets transport variability, it can impact the mean values slightly. With 166 glider transects over this period, we consider the mean transport estimate of $1.0\pm0.3$ Sv derived from glider data to be robust. This is supported by agreement within uncertainties with independent estimates from the previous methodology, which remained consistent across observational periods: $1.5\pm0.2$ Sv for 2020–2023 in this study, $1.3\pm0.2$ Sv for 2014–2020 in Fraser et al. (2022), and $1.4\pm0.3$ Sv for 2014–2018 in Houpert et al. (2020). Although uncertainties overlap, the previous method yields higher mean ESC transports than the new approach, particularly in the upper 800 m above the southward undercurrent (Figure 8). This underscores the importance of glider data for resolving the spatial structure of the ESC in the new reconstruction (Figure 4).

Higher-frequency variability with periods between 23 and 120 days is elevated in the new transport reconstruction compared to the old one (Figure 7). These periods align with typical mesoscale activity in the Rockall Trough (Ullgren and White, 2012). This enhancement is likely due to the improved spatial representation of mesoscale processes introduced through EOF mode 2 (Figure 3c), which also enables a better depiction of extreme transport events (Figure 6a).

For the seasonal cycle, glider data remain sparse and do not cover every month (Figure 6c,d). The new transport reconstruction shows a weak seasonal cycle compared to the monthly standard deviation for 2020–2023, with minima in January and August and elevated northward transport between March–July and September–November. This largely agrees with the findings of Fraser et al. (2022) based on glider data for 2020–2022. However, the extended 2020–2023 glider transport in this study does not show the August minimum, highlighting the importance of a continuous ESC transport product. The seasonal cycle based on the new reconstruction contrasts with Xu et al. (2015), who found a simple cycle peaking in winter and dropping in summer based on geostrophic ESC estimates referenced to satellite data. This underscores the importance of in situ observations for realistically capturing ESC variability.

A limitation of the new methodology is related to the available data coverage. When the EB1 observations and GLORYS output at the RTADCP location do not capture the main features of an EOF mode—such as being located in an area of minimum amplitude—the corresponding regression coefficient will be small, and the mode cannot be reproduced accurately. In such cases, including higher-order modes does not improve the reconstruction and may introduce noise (Figure 3). Therefore, only modes whose dominant features are represented at the mooring locations contribute meaningfully to the transport estimates.

## 4.2 Part II – Full Rockall Trough transports 2014-2024

Mean volume transport agree well with previous Rockall Trough transport estimates based on shorter time periods (Houpert et al., 2020; Fraser et al., 2022). Total Rockall Trough transport is dominated by the NAC flowing through the mid basin, with the ESC acting as a secondary contributor in the eastern wedge and a negligible contribution from the southward flow through the western wedge, which is anti-correlated with the NAC (Figure 1 and 9). For the first time, we present heat and freshwater transports derived from moored observations. We tested the effect of using only temperature and salinity profiles at the mooring locations in ship-based and glider-based hydrographic and velocity sections and found them to provide a good approximation (Table S1). The mooring-derived heat and freshwater transports are largely dominated by volume transport (Table 1), in good agreement with Gary et al. (2018), who investigated Rockall Trough volume, heat and freshwater transport using hydrographic sections along the Extended Ellet Line.

Heat and freshwater transport estimates are sensitive to the choice of reference temperature and salinity. In this study, temperature and salinity are referenced to the depth-averaged temporal mean of the western moorings over the full observation period. Through this choice, the resulting heat and freshwater transports for the mid basin and eastern wedge primarily reflect variability associated with the NAC and ESC within the northern Rockall Trough. This differs from the approaches used for the OSNAP array (Fu et al., 2025) and for the NAC branch in the Iceland Basin (Dotto et al., 2025), which adopt a more basin-wide perspective. In particular, both studies reference heat transport to the in situ freezing point, which leads to substantially larger northward heat transport estimates than those obtained using our local reference values.

The dominant driver of interannual NAC variability in the mid basin, which also dominates total Rockall Trough transport variability, is changes in temperature (Figure 10). Total transport decreased significantly by $-0.22$ Sv/year over the last 10 years, which we interpret as a remnant of multi-year hydrographic changes associated with the subpolar cold freshwater anomaly (Holliday et al., 2020; Fox et al., 2022). Overall, temperature changes caused a weakening of the mid basin transport of $-0.52$ Sv/year before 2022, partly offset by a salinity-driven strengthening of $0.21$ Sv/year. In the most recent year, positive temperature and salinity anomalies in the Rockall Trough (Figure S4) acted together to strengthen the mid basin transport by $0.85$ Sv/year. Subsurface hydrographic anomalies began to develop in 2022, with the strongest signal coinciding with the arrival of the basin-wide extreme North Atlantic marine heatwave in 2023 (Berthou et al., 2024; England et al., 2025). The heatwave is surface-intensified but its signal extends through nearly the full water column in both the eastern and western Rockall Trough and shows little decay by the end of the moored observations (Figure S4). Please note, while volume transport dominates, temperature (salinity) anomalies may amplify or dampen heat (freshwater) transport variability, especially in the mid basin for freshwater and western wedge for heat transport (Table S3).

ESC variability acts as a secondary driver of total Rockall Trough variability and shows no significant trend between 2014 and 2024. We find higher EKE west of EB1 during periods of high ESC transport, accompanied by a stronger meridional (approximately along-slope) wind stress component. This aligns with previous studies showing that a stronger ESC is associated with stronger along-slope winds (Huthnance, 1984; Marsh et al., 2017). The elevated EKE may reflect an intensified slope current enhancing boundary vorticity. Most low ESC transport events occur during winter, when the mixed layer is deeper and the ESC flow is predominantly barotropic (Marsh et al., 2017). Low ESC transport is also associated with high-amplitude wind stress and wind stress curl anomalies in the eastern subpolar North Atlantic, characterised by a strong westward wind component (Figure 11). Winter storms often have a pronounced westward component, which tends to disrupt the slope current and enhance ocean–shelf exchange (Jones et al., 2020). During these low-transport periods, local EKE is reduced, consistent with piloting experience that shows noticeably less deflection of glider transects by strong currents and eddies in winter.

Both the NAC and ESC were linked to larger-scale multi-year variability in the North Atlantic, including changes in the extent of the subpolar gyre and anomalous wind patterns associated with the North Atlantic Oscillation in previous studies (Häkkinen and Rhines, 2004; Hátún et al., 2005; Marsh et al., 2017). While the ESC shows no clear multi-year changes or long-term trend, our results reveal pronounced multi-year variability in the NAC transport. Prior to 2017, the eastward extension of the subpolar gyre associated with the North Atlantic cold freshwater anomaly is linked to stronger transport in the Rockall Trough NAC branch and reduced NAC flow in the Iceland Basin (Dotto et al., 2025). Between 2017 and 2022, Dotto et al. (2025) reported a stronger NAC in the Iceland Basin associated with a westward contraction of the subpolar gyre, consistent with the weakening of the NAC in the Rockall Trough observed in this study. This multi-year variability is most apparent in the freshwater transport through the Rockall Trough mid basin, with enhanced northward salt transport before 2017 and after 2022, corresponding to more negative freshwater transport (Figure 9e–f).

The longer-term NAC transport variability agrees well with meridional overturning estimates from OSNAP in the Rockall Trough (Fu et al., 2025), highlighting that the total flow through the Rockall Trough participates in the upper limb of the subpolar overturning circulation. The multi-year anti-correlation between the NAC branches in the Rockall Trough and the

Iceland Basin does not necessarily hold on shorter interannual timescales; for example, in mid-2021 the entire NAC exhibited low transport (Figure 9; Figure 8a in Dotto et al., 2025). Resolving subpolar North Atlantic NAC variability therefore requires accounting for all branches from the Rockall Trough to the Iceland Basin. With both NAC transport products now available for a decade, a combined analysis to characterise full NAC variability along the OSNAP line is the logical next step.

## 5 Conclusion

We present a decade-long record of Rockall Trough circulation and property transports from the Ellett Array (2014–2024), extending previous volume transport estimates (Houpert et al., 2020; Fraser et al., 2022) and introducing the first mooring-based heat and freshwater transport calculations. For the first time, we integrate glider observations with mooring and reanalysis data to reconstruct eastern boundary transport, enabling more accurate estimates that realistically capture key features such as the ESC, the southward undercurrent, and mesoscale variability.

Rockall Trough transport variability is dominated by the NAC in the mid-basin, with multi-year changes linked to subpolar gyre dynamics, the decay of the mid-2010s North Atlantic cold freshwater anomaly, and recent warming amplified by the 2023 North Atlantic heatwave. The ESC acts as a secondary contributor, with variability driven primarily by along-slope wind stress rather than large-scale multi-year changes in the North Atlantic.

This study highlights the value of targeted observing systems for delivering accurate, continuous records of ocean circulation required for Atlantic climate monitoring. Gliders provide unique insights into boundary currents that are difficult to obtain using traditional platforms. When integrated with mooring data using the methodology presented here, they become a powerful and complementary component of the Ellett Array observing system. Our approach can be directly applied to the wider challenge of monitoring ocean transport using heterogeneous combinations of mooring and glider platforms.

*Code and data availability.* Code available on github: https://github.com/ScotMarPhys/Rockall_Trough_Transports/tree/v2024.0. Rockall Trough data available on: https://thredds.sams.ac.uk/thredds/catalog/osnap/catalog.html. Global Ocean Physics Reanalysis GLORYS12V1 were obtained from the Copernicus Marine and Environment Monitoring Service (CMEMS) Marine Data Store (MDS): https://data.marine.copernicus.eu/product/GLOBAL_MULTIYEAR_PHY_001_030/description, DOI:10.48670/moi-00021 (Accessed on 03-Oct-2025). Altimetry data were obtained from CMEMS Marine Data Store (MDS): https://data.marine.copernicus.eu/product/SEALEVEL_GLO_PHY_CLIMATE_L4_MY_008_057/description, DOI: 10.48670/moi-00145 (Accessed on 01-Mar-2023). ERA5 monthly averaged data on single levels were obtained from Copernicus Climate Change Service (C3S) Climate Data Store (CDS): https://cds.climate.copernicus.eu/datasets/reanalysis-era5-single-levels-monthly-means?tab=overview, DOI: 10.24381/cds.f17050d7 (Accessed on 05-Apr-2024). GEBCO bathymetry data version 20141103 were obtained from: https://www.gebco.net/data_and_products/gridded_bathymetry_data/version_20141103/.

*Author contributions.* KB planned and performed the analysis. KB, SCJ and NJF authored the paper. LAD, SCJ and KB post-processed the mooring and glider data. SAC, MEI and NPH secured funding for the research. All co-authors contributed to the scientific improvement of the paper.

*Competing interests.* The contact author has declared that none of the authors has any competing interests.

*Disclaimer.* This output reflects only the author's view, and the European Union cannot be held responsible for any use that may be made of the information contained therein.

*Acknowledgements.* We thank the captains, crews, scientists, and technical groups involved in the different national and international research cruises, on research vessels RRS Discovery, RRS James Cook, RRS James Clark Ross and RSS Charles Darwin to the subpolar North Atlantic for their contributions in collecting CTD, velocity, and mooring data and for making them freely available. This project was supported

by the UK Natural Environment Research Council National Capability programme AtlantiS (NE/Y005589/1), NERC Grants UK OSNAP (NE/K010875/1 and NE/ K010875/2), UK OSNAP Decade (NE/T00858X/1) and ODISSEA (NE/Y005236/1). This project has received funding from the European Union's Horizon 2020 research and innovation programme under grant agreement No. 818123 (iAtlantic).

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
