# Peer review of "A decade of continuous Rockall Trough transport observations using moorings and gliders"

_EGUsphere, 2025_

## Author Comment (AC1)

**Response to Reviewer 1 for "Eight years of continuous Rockall Trough transport observations using moorings and gliders"**

We thank the reviewers for their constructive comments which help us improve the quality of our manuscript. Below, we provide detailed responses to each comment.

During the revision, we are implementing the following improvements to the transport calculations, which we are making independently to strengthen the analysis:

- **Extending the mooring dataset to 10 years**, now overlapping with the entire glider observation period. We are changing the title of the manuscript to reflect this update: "A decade of continuous Rockall Trough transport observations using moorings and gliders"

- **Correcting the EOF analysis** by using the original time steps of glider transects instead of 15-day averages, which previously included an irregular number of transects.

- **Correcting an error in the new methodology:** The EOF analysis and regression are applied to velocity anomalies. In the earlier version, we mistakenly subtracted the glider mean at EB1 and RTADCP positions from EB1 data and GLORYS12V1 output instead of subtracting the mean of each respective dataset. This introduced a systematic offset, which has now been corrected. The glider mean field is added at the final step to define the mean of the reconstructed section, eliminating the need for bias correction of the GLORYS12V1 output.

Reviewer 1:

Burmeister et al. present a new method of resolving spatiotemporal differences in sampling between moorings and gliders at the Rockall Trough sector of the Ellett Array. This region is one of repeat monitoring and an important region for North Atlantic circulation. With greater integration of autonomous platforms (gliders, AUVs) to traditional repeat monitoring methods (fixed moorings), the authors are tackling a very relevant problem to the community. While the science is worthwhile, we have concerns about the paper itself and suggest major revisions are needed before this manuscript can become a suitable paper.

Major comments:

The overall structure of the paper is inconsistent and confusing, taking focus from the science and making it very difficult to read. The paper lacks well defined methods, results, and discussion sections. Throughout the paper, the authors seem to jump between methods, results, and discussions no matter the actual location in the paper. I would highly recommend the authors follow a more traditional paper structure.

Thank you for highlighting that the structure of the paper can be improved for clarity. We are revising the entire paper accordingly.

The paper relies extremely heavily on a previous paper from the second author (Fraser et al., 2022). The paper doesn't seem to sufficiently introduce this paper in a clear manner. Currently, to understand the full context of Fraser et al., and how their method differs from this previously published method, the reader needs to go and seek out Fraser et al. Additionally, we noticed some portions of the text between the two papers was very, very similar with only a word or two of difference between the two. We suggest that the authors take care not to copy themselves.

Thank you for pointing this out. We are revising the section for clarity. While the methodology is published in detail in Fraser et al. (2022), we recognise that it is relevant to the theme of this special issue. To support accessibility and transparency, we are adding more detailed methodological information in the supplementary information. Please note that this study extends the dataset and approach presented in Houpert et al. (2020) and Fraser et al. (2022). As such, some overlap in the methods section is unavoidable due to the use of consistent techniques. Nonetheless, we are taking care to ensure that the presentation is clear and appropriately contextualised.

Please keep consistency in grammar, capitalization, etc. throughout the paper. For example, figure captions are sometimes "longitude", "lon", "Longitude [degE]", or "Longitude [ºE]".

Thank you for highlighting. We are editing all text and figures accordingly.

The authors only briefly mention an important difference between the model and observation output; the model does not capture the extreme events that the glider does. In a region like the North Atlantic that is known to be physically dynamic (Holliday et al., 2006; Johnson et al., 2024), this difference is noteworthy and the implications of such should be discussed more. Can you explore the implications of this more? One method could be to calculate monthly and annual budgets with and without the extreme events to better understand the net impact they have?

Thank you for highlighting this important point. We agree that the difference between model output and glider observations, particularly regarding extreme events, is noteworthy in such a physically dynamic region. During the revision, we identified an error in the application of

our methodology: prior to regression, the mean of the glider at EB1 and RTADCP location had been incorrectly subtracted from both the EB1 data and GLORYS12V1 output. After correcting this, the revised approach now reproduces extreme events more realistically (Figure 1) by incorporating the second EOF mode, which we interpret as representing mesoscale eddies. This improvement significantly enhances the fidelity of the reconstructed fields. We are updating the results and discussion sections to emphasise the implications for extreme events and their potential impact on transport estimates.

[Figure]

*Figure 1: a) Time series of volume transport for the upper 1000 m derived from meridional velocity section based on glider transects (blue line), the new eastern wedge reconstruction (EW new, orange line), the old eastern wedge reconstruction (EW old, black dotted line). (b) Transport time series of the new approach averaged ±1 day on the time step of the glider observations (EW new resampled, orange dashed line). Glider transport are shown in blue again for reference. (c) Monthly mean seasonal cycle for glider (blue line), the new eastern wedge reconstruction (EW new, orange line) and the resampled new eastern wedge reconstruction (EW new resampled, orange dashed line). The error bars mark ±1 standard deviation. (d) Number of glider transects per month of the year.*

The Discussion reads like a conclusion, and much of what should be included in the Discussion is throughout. Currently, the paper ends abruptly and the "why should you employ this method" is missing.

Thank you for highlighting that the manuscript does not clearly convey the benefits of the new methodology. We are revising the discussion to emphasise why this approach should be employed. The main advantages are:

- Improved accuracy of the mean strength and structure of the ESC, based on multiyear glider observations rather than a bias correction of GLORYS12v1 data using only eight months of ADCP measurements.

- Enhanced ability to reproduce extreme events, likely associated with mesoscale variability, through inclusion of the first two EOF modes.

- High-resolution ESC product in both space and time, reducing aliasing effects caused by temporally scattered glider data.

We are editing the text throughout the manuscript to clarify these points and ensure that the discussion section clearly communicates the value of this methodology.

Line by line comments:

Line 13: What is European temperature? Temperature of Europe?

We are editing the entire introduction section and this phrase is no longer part of it.

Line 15: Does order one mean "first order"? Can the authors clarify what they mean by this sentence?

Thanks for highlighting. We are editing the text for clarity.

Line 31: Presumably some of the observational difficulty in this region stems from ship traffic. Has there been any effort to integrate commercial shipping data?

No, there has been no effort to integrate commercial shipping data into our observations for this region.

Line 31-32: I would suggest putting part about RTADCP into methods and add abbreviation definition

Thank you for pointing this out. We are editing the manuscript accordingly.

Line 40: Perhaps change "amassed" to "completed"?

We are editing the entire introduction section and this phrase is no longer part of it.

Line 41: The reader needs more information on Fraser et al method without having to chase it down themselves.

Thank you for highlighting. As mentioned above we are adding further information about the methodology in the supplementary information.

Line 56: Please stay consistent with "Seaglider" or "glider".

Thank you for pointing this out. We are editing the manuscript accordingly.

Line 69-70: Can you expand on this with how? I can see it is discussed below, but the how should come in the first sentence.

Thank you for your comment. Unfortunately, it is not entirely clear to us what the reviewer is referring to. The sentence in question introduces the methodology by Fraser et al. (2022), and the detailed explanation of how this method works is provided in the subsequent section.

Line 70: The sentences "glider transects are invariably affected by the ocean velocity field and hence follow irregular and inconsistent trajectories. The different transects do not correspond spatially and do not in general have the same cross-sectional area" are a bit too close to Fraser et al. 2022: "However, glider transects are invariably affected by the ocean velocity field and hence follow irregular and inconsistent trajectories. As a result, different transects do not correspond spatially and, due to the variability in slope shape and steepness at different locations, do not in general have the same cross-sectional area."

As mentioned above, this study builds on the dataset and methodology presented in Houpert et al. (2020) and Fraser et al. (2022). Given the continuity in approach, some textual overlap in the methods section is unavoidable. However, we are reviewing the manuscript carefully to ensure that all reused descriptions are necessary, appropriately cited, and clearly contextualised within the scope of this study.

Line 76: Can you expand on what you mean by the correct isobath? Is this equivalent to essentially depth binning

Thank you for pointing out that this section lacked clarity. We are revising the text to better explain that data from individual glider transects are allocated to coordinates along the standard transect that share the same isobath.

Figure 1b : Please make the font larger.

Figure 1b: The black cross hatching is very difficult to see.

Figure 1b: What do the green triangles, blue boxes, and red circles represent? Please be explicit.

Figure 1a: Please remove the abbreviations if they are not mentioned.

Figure 1a: This panel is missing the 'A'.

Thank you for your five suggestions regarding Figure 1. We are revising Figure 1 and its caption for completeness, clarity, and consistency.

Line 104: What does "generally very high" mean? Please be explicit.

Thank you for highlighting the need for greater clarity. At EB1, 78% of CTD data and 89% of current meter data were successfully recovered. At WB1 and WB2, 85% of CTD data were recovered, along with over 99% of current meter data at WB1 and 97% at WB2. We are adding these details to the manuscript.

Line 107-108: This statement is redundant with the introduction and can likely be removed.

We used the RTADCP to correct the GLORYS12V1 output. We are editing the paragraph for clarity.

Line 105: Typo- "Gap filling".

Thank you for pointing this out. We are correcting the typo.

Figure 2: If you are going to include cruise IDs, those cruises need to be listed somewhere. Perhaps a table in the supplemental?

Figure 2: Please increase the font size.

Thank you for your two suggestions regarding Figure 2. We are revising Figure 2 and its caption for completeness, clarity, and consistency.

Line 109: Do you mean "auxiliary data"?

Thank you for pointing this out. We are correcting the section title accordingly.

Figure 2: Does the y-axis represent a fixed pressure? Why does this differ from the black lines when they are supposed to be the same variable?

The black lines represent the actual instrument depth over time, while the shading shows interpolated fields on a regular depth grid. Both appear in the same plot. Please note that we are converting pressure to depth for consistency across all figures in the manuscript.

Figure 3: Please define EOF in the caption.

Thank you for your suggestions. We are revising Figure 3 and caption for completeness, clarity and consistency.

Line 119-121: I appreciate the comparison of this "new" method to the previous method.

Thank you.

Line 125-126: I would suggest you contrast your method with the previously utilized method of Brandt.

Thank you for the suggestion. Brandt et al. (2014, 2016, 2021) use a combination of moored and ship-based observations to estimate the transport of the Atlantic Equatorial Undercurrent. Instead of standard EOF patterns, they apply HEOF (Hilbert Empirical Orthogonal Function) patterns to better capture the vertical displacement of the current core. In our study, we evaluate both EOF and HEOF approaches with similar results and choose to use EOF patterns for simplicity. We are adding further explanation in the text to clarify this choice.

Figure 4: This is a very nice figure.

Thank you.

Figure 4: Is 'ADCP' 'RTADCP'? Please be consistent .

Thank you for your suggestions. We are revising Figure 3 and caption for completeness, clarity and consistency.

Lines 138-139: Please expand on this EOF analysis more.

Thank you for indicating that more information is needed for clarity. We are moving this part into the data and method section and are adding additional information about HEOF accordingly.

Line 144: Should GLORYS2v12 be GLORYS2v1?

Thank you for pointing this out. For simplicity we are referring to the ocean reanalysis now as GLORYS throughout the manuscript and introduce the version in the data and method sections accordingly..

Equations 1 and 2: Shouldn't this whole section be in a better-defined methods section?

Thank you for highlighting this point. Section 3 on transport calculations is now being incorporated into the data and methods section for clarity and consistency.

Lines 149-154: Can you add significance tests to these correlations to further validate the approach?

Thank you for this suggestion. All regression results are statistically significant, and we are adding the corresponding details to the manuscript for clarity.

Line 158-159: Can Section 3 be restructured so that "the volume, heat and freshwater transports are then calculated from the reconstructed eastern wedge velocity and hydrography sections using Equations 3-5" immediately precedes these calculations?

Thank you for your suggestions. We are revising the section accordingly.

Line 157-158: It does not look like that reconstructed data is present in Fig 4b? Did you mean above 1000 m?

The reconstructed data below 1000 m is included; however, it may appear absent because the temporal mean velocities at EB1 below this depth are close to zero.

Line 157-158: What are the upper and lower bounds of depth bins?

Thank you for highlighting a lack of clarity here. As mentioned in Section 2.2, all mooring data are vertically gridded onto a regular 20-dbar grid. We are adding this information to the section describing the new eastern wedge reconstruction for clarity.

Section 3.2: Much of this section could better fit in a proper Discussion section.

Section 3.2 describes the previously published methodology used to reconstruct the eastern wedge velocity section as presented in Houpert et al. (2020) and Fraser et al. (2022). We have retained this description because the approach is used for comparison purposes and remains relevant, particularly since the new methodology incorporates certain elements of the original method.

Lines 193-195: I find these sentences unnecessary, but that is personal preference.

Thank you for your feedback. As part of the revised structure, we are splitting the results section into two parts for clarity. We are keeping these sentences because they help introduce the organisation of the section, and we are editing them for improved clarity.

Figure 5b: Please fix formatting of the legend so it does not cover part of the figure.

Figure 5: Please differentiate between a and b in the legend.

Figure 5: Somewhere, please mention/explain the glider data gaps.

Figure 5: Can you put a and b y-axes on the same scale?

Figure 5: Maybe chose different colours to make it more obvious when glider data is missing?

Thank you for your 5 suggestions regarding Figure 5. We are revising Figure 5, caption and text for completeness, clarity and consistency.

Figure 6: This is a nice figure.

Thank you.

Line 185: Can you provide the reference density and specific heat capacity?

Thank you for pointing this out. We are editing the manuscript accordingly.

Line 208: Can you please expand on the bias correction?

Of course. A bottom-mounted Acoustic Doppler Current Profiler (RTADCP) was deployed in the ESC core in 2014, but only eight months of data were recovered; later deployments failed due to damage, likely from fishing (Houpert et al., 2020). To compensate, Houpert et al. (2020) used GLORYS12V1 meridional velocity fields, which captured variability but underestimated flow strength by about +7.6 cm/s. They corrected this bias by applying a uniform offset of +7.6 cm/s at the RTADCP location (57.1°N, 9.3°W, upper 750 m). We are adding these details to the data and methods section for clarity.

Lines 242-249: Comparisons between Fraser et al and this study should be better integrated.

Thank you for highlighting this point. We are adding a paragraph in the discussion section that compares the previous methodology with the new approach.

Lines 251-255: The discussion does not need to begin with a recap of the study. This is redundant and would be better fitted as a single sentence in the start of the conclusion.

Thank you for highlighting. As part of the revision, we are restructuring the entire manuscript and are introducing separate sections for discussion and conclusion to improve clarity and readability.

Figure 7: Please specify what red and blue shading are in the caption.

Thank you for pointing this out. We are editing the manuscript accordingly.

Line 263-264: Throughout, the authors will make statements like: ". . .declining northward transport in the mid basin was counteracted by decreasing southward flow at the western boundary" but it would be good to have a figure showing these type of spatial trends.

Thank you for this suggestion. We are adding a table in the supplementary information listing the trends fitted to the different transport estimates.

Lines 269-270: If the ESC is "disproportionally important for poleward heat and freshwater fluxes", it should be discussed more, and its importance should be brought up earlier in the paper. Is this from the literature or a finding?

Thank you for highlighting this point. The statement refers to findings from previous studies (Clark et al., 2022; Daly et al., 2024), which emphasise the importance of the ESC for on-shelf heat and freshwater transports. The term "disproportionally" reflects its relatively small transport compared to the NAC in the Rockall Trough; however, NAC transport in this region does not significantly affect European shelf exchanges, whereas the ESC does. We are editing the sentence for clarity.

---

## Author Comment (AC2)

**Response to Reviewer 2 for "Eight years of continuous Rockall Trough transport observations using moorings and gliders"**

We thank the reviewers for their constructive comments which help us improve the quality of our manuscript. Below, we provide detailed responses to each comment.

During the revision, we are implementing the following improvements to the transport calculations, which we are making independently to strengthen the analysis:

- **Extending the mooring dataset to 10 years**, now overlapping with the entire glider observation period. We are changing the title of the manuscript to reflect this update: "A decade of continuous Rockall Trough transport observations using moorings and gliders"

- **Correcting the EOF analysis** by using the original time steps of glider transects instead of 15-day averages, which previously included an irregular number of transects.

- **Correcting an error in the new methodology:** The EOF analysis and regression are applied to velocity anomalies. In the earlier version, we mistakenly subtracted the glider mean at EB1 and RTADCP positions from EB1 data and GLORYS12V1 output instead of subtracting the mean of each respective dataset. This introduced a systematic offset, which has now been corrected. The glider mean field is added at the final step to define the mean of the reconstructed section, eliminating the need for bias correction of the GLORYS12V1 output.

**Recommendation**:

Requires major revision before acceptance to "Ocean Science". Suggested improvements and key points are listed below.

**Main points requiring revisions:**

This study presents an updated methodology for analysis of observational data from Rockall Trough. The new methodology is supposedly better than existing methods, but there is no hard evidence that the new version is indeed an improvement. I recommend adding analyses that quantitatively show why this is the case. To do this, the following approaches come to my mind (but the authors may choose differently):

- One could test the different methods in a numerical model that has sufficient resolution and fidelity in the Rockall Trough (validate the model first - does not need to be GLORYS).

In the model world, the actual flow (of volume, heat, and freshwater) is known, and the observing system results can be simulated (in an OSSE type analysis).

Thank you for these suggestions. Houpert et al. (2020) performed a comprehensive error analysis. We are adding a subsection in the data and method section of the manuscript. Additional details are provided in the expanded comments below.

- If the biggest improvement is the better time resolution compared to earlier methods, one could also compare spectra of the resulting time series from old/new methods. If the new method is really superior, the shape and power levels in the spectra should demonstrate what kind of aliasing is now avoided, and what the presumably lowered noise floor at high frequencies looks like.

Thank you for highlighting this point. We would like to clarify that both the old and the new methodologies for reconstructing the eastern section use the same temporal resolution, based on mooring data and GLORYS12v1 output at the RTADCP location.

Since 2020, we have additional glider observations that provide detailed insight into the structure of the European Slope Current (ESC). However, these observations remain scattered in time and are therefore prone to temporal aliasing. For the first time, mooring and glider observations overlap, allowing direct comparison. We find that the old method agrees well with glider transports, but it fails to realistically reproduce extreme events, which are likely driven by enhanced mesoscale activity.

During this review, we identified an error in the new approach. After revision, the updated method now successfully reproduces these extreme events (see Figure 1, page 3 in our response to Reviewer 1) by including the second EOF mode, which we interpret as representing mesoscale eddies. We agree with the reviewer that a spectral comparison of the time series would provide valuable additional information, and we are including this analysis in the revised manuscript.

The study does not quantify uncertainties in the resulting time series. Error bars, in the few places where they exist, denote the error of the mean when averaging e.g. several years into an annual cycle, but this says nothing about the uncertainty of the original time series itself. Can we believe the smaller wiggles in figure 5a, or are they instrumental noise?

Houpert et al. (2020) assessed the error arising from the horizontal extrapolation of the current meters and reliance on ocean reanalysis at the eastern wedge against lADCP data from cruises along the section. For the EW transports, they found a total mean bias error for the method of -0.21 Sv and a Root-Mean-Square Error (RMSE) of 0.59 Sv (see their SI text S1.3, Table S1, Figure S6 and S8). As mentioned above, we are adding information about the transport accuracy in the data and method section of the manuscript.

Is the ~0.8 Sv misfit (l. 150) the dominant contribution to uncertainty, or is it the sensor errors that are listed for glider and mooring CTD - how many Sv result from these sensor errors?

Thank you for raising this point. The ~0.8 Sv misfit was primarily caused by an error in the previous implementation of our methodology rather than by sensor inaccuracies. Specifically, the EOF analysis and regression were applied to velocity anomalies, and the glider mean field was added at the final step to define the mean of the reconstructed section. This approach eliminates the need for bias correction of the GLORYS12V1 output.

The misfit occurred because, in the earlier version, we mistakenly subtracted the glider mean at EB1 and RTADCP positions from EB1 data and GLORYS12V1 output instead of subtracting the mean of the respective datasets themselves. This introduced a systematic offset, which we have now corrected. After this revision, the updated method aligns well with glider observations and no longer exhibits the previous misfit.

I recommend including a systematic accounting of the major sources of uncertainty, and adding those to the relevant figures and results. Just copying the manufacturer's sensor specifications and not propagating them into the resulting volume, heat, and freshwater fluxes is, in my opinion, insufficient.

Thank you for these suggestions. Houpert et al. (2020) conducted a comprehensive error analysis of the transport estimates using a Monte Carlo approach to assess the impact of instrument errors, and evaluated methodological uncertainties using LADCP and CTD data from the Ellet Line hydrographic sections, alongside climatology data from MIMOC (Schmidtko et al., 2013).

In the mid basin, the Monte Carlo simulations showed the combined effect of pressure, temperature, and salinity resulted in a RMSE of 0.05 Sv. For the western wedges, current meter inaccuracy led to a maximum transport error of ±0.12 Sv at a 68% confidence level.

Methodological errors were assessed by comparing transport estimates derived from full LADCP/CTD sections with those derived from subsampled mooring data using the same processing methods. Errors were attributed to gridding and vertical extrapolation for the mid basin (bias error of 0.11 Sv, RMSE of 0.34 Sv for sull data return), gridding and horizontal extrapolation for the western wedge (bias error of -0.30 Sv, RMSE of 0.63 Sv) and horizontal extrapolation and reliance on ocean reanalysis for the eastern wedge (bias error of 0.21 Sv, RMSE of 0.59 Sv).

Mean bias and RMSE for the total transport were obtained by combining the errors of all three subsection. For optimal data return, the total RMSE was 0.93 Sv and the mean bias error was 0.03 Sv. However, data loss due to instrument failure in 2014-2015 and 2016-2017 increased uncertainty in the mid basin transports resulting in a bias error of -0.39 Sv and 0.38 Sv as well

as a RMSE of 1.10 Sv and 0.93 Sv, respectively. For further details, we refer readers to Houpert et al. (2020). We are adding a subsection about the error estimates in the data and method section in the manuscript.

The methodology for heat and freshwater transports is flawed in that underlying data are not available at sufficient spatial resolution, let alone with appropriately co-located measurements. As a bare minimum, there needs to be a validation why the method should still give correct results. This could (again) be done with an OSSE-type numerical simulation. If these validations have been done in some of the referenced work, they need to be summarized here. I did not review the section with heat and freshwater transports at this time.

Thank you for your suggestion. To assess whether mooring-based hydrographic profiles are sufficient for estimating heat and freshwater transports, we compare transports derived from full ship and glider sections with those calculated using ship and glider profiles subsampled at mooring positions (Table 1). For the western and eastern wedges, mean bias errors were small (1–6% of the mean transport), and RMSE values were well below one standard deviation (4–8%), except for freshwater transport in the eastern wedge (19%). For the mid basin, errors were larger (bias: 22–31%; RMSE: 25%), but given the high natural variability, we consider these results as acceptable. Full details are provided in Table 1 and we are adding a subsection in the data and method section of the manuscript.

*Table 1: Comparison of heat ($Q_h$) and freshwater ($Q_f$) transports estimated from full temperature–salinity sections versus profiles at mooring positions. "Mean full" and "Std Dev full" represent the mean and one standard deviation calculated from complete ship sections (western wedge, mid basin) or full glider sections (upper 1000 m, eastern wedge). "Mean profile" and "Std Dev profile" represent calculations using data only at WB1/2 and EB1 positions from ship sections (western wedge, mid basin) or glider data (upper 1000 m, eastern wedge). Mean bias error and root-mean-square error (RMSE) between full-section and profile-based heat and freshwater transports are also shown. Only ship sections covering all Extended Ellet Array stations in the western wedge or the mid basin, respectively, were used.*

| | Western wedge 10 ship sections | | Mid basin 8 ship sections | | Eastern wedge 166 glider sections | |
|---|---|---|---|---|---|---|
| | $Q_f$ ($10^{-2}$ Sv) | $Q_h$ ($10^{-2}$ PW) | $Q_f$ ($10^{-2}$ Sv) | $Q_h$ ($10^{-2}$ PW) | $Q_f$ ($10^{-2}$ Sv) | $Q_h$ ($10^{-2}$ PW) |
| Mean full | 0.23 | -0.43 | -2.79 | 6.20 | -0.43 | 1.19 |
| Mean profile | 0.24 | -0.42 | -3.66 | 7.59 | -0.44 | 1.12 |
| Mean bias error | -0.01 | 0.00 | 0.86 | -1.39 | 0.01 | 0.07 |
| Std Dev full | 1.57 | 3.33 | 5.62 | 11.04 | 0.66 | 1.67 |
| Std Dev profile | 1.62 | 3.42 | 5.12 | 10.13 | 0.67 | 1.60 |
| RMSE | 0.08 | 0.12 | 1.46 | 2.90 | 0.12 | 0.14 |

There is an inconsistency or flaw in the method, in that it uses EOF patterns but use of patterns higher than order one fails to improve the results in the validation step. I suspect this is due to how observational and model data are mixed in the methodology. There are comments about this below, as well as suggested steps to address this.

Thank you for this observation. We do not consider this a flaw in the methodology but rather a limitation related to the available data coverage. When the subsampled data (EB1 observations and GLORYS12V1 output at the RTADCP location) do not capture the main features of an EOF mode—such as being in an area of minimum amplitude—the corresponding regression coefficient will be small, and the mode cannot be reproduced accurately. In such cases, including higher-order modes does not improve the reconstruction and may introduce noise. Therefore, only modes whose dominant features are represented at the mooring locations contribute meaningfully to the transport estimates.

Our approach is designed to minimise reliance on model output, using it only where observational gaps cannot otherwise be filled. We are adding a paragraph in the discussion section to clarify this limitation and provide additional context in the comments below.

The overall presentation needs to be polished; I am including detailed comments below. For a publication-ready manuscript, I expect more consistency with labels, abbreviations, etc., across the figures and text.

Thank you for pointing this out. We are revising all figures and text to ensure clarity and consistency throughout the manuscript. This includes harmonising labels, abbreviations, and formatting across figures and captions, as well as improving readability in the main text.

**Improvements to content:**

Ll. 47-49:

In the transition between the paragraphs here, it is not clear whether this study solves the shortcomings of the glider observations, or reveals what they were in the first place. Maybe change the wording in the paragrph lines 49 ff. to make this clear.

Thank you for pointing this out. We are revising the entire introduction for clarity.

Ll. 47 and 85:

These state that the glider data are "scattered and sparse in time". It is not very clear what this means, nor what would be considered "good enough". Can this be clarified/quantified somehow? For the introduction, this is probably OK as is, but in the methodology section, I recommend being more quantitative.

Thank you for pointing this out. We are revising the text for clarity.

Ll. 61 ff.:

The glider CTD sensors are essentially the same as the mooring ones. You are listing what appears to be the factory specifications for sensor accuracy here. For the moorings described later, you list similar accuracies but presumably requiring the tedious calibration procedures referenced there (line 97; McCarthy et al. 2015). Are the glider data processed with similar methods? If not, they will not be as accurate as described.

Thank you for your comment. The accuracy values provided refer to manufacturer specifications and differ from those listed later for the moorings. Glider CTD sensors and compasses were calibrated in the laboratory before each mission, and an in-water compass calibration was performed at the start of each deployment. However, the processed glider data do not achieve the same accuracy as the fully calibrated mooring sensors. Fraser et al. (2022) found that glider temperature profiles at EB1 agree well with moored profiles, while salinity is consistently underestimated by approximately 0.02 g/kg. Importantly, Fraser et al. (2022) also showed that this salinity bias has minimal influence on geostrophic velocity shear and associated transport estimates. We are editing the text for clarity.

Ll. 93-95:

It is not clear how the WB1 and WB2 moorings are concatenated. The reference quoted in turn refers to another reference (McCarthy et al., 2015), which also leaves details somewhat open. There is a "correct" way to do this merger, assuming geostrophy and that the current and CTD data are actually available: One can start by integrating the CTD-derived specific volume anomaly upwards from the bottom of the deep mooring. Then, one can "jump" horizontally to the shallower mooring using the geostrophic equation and the currents at the depth of the jump (supposedly some average from nearby current meters on both moorings can be used). Then, continue integrating the CTD-derived data up along the shallower mooring. Is this what was done?

Thank you for pointing out we referencing to the wrong paper. We use the approach of Fraser et al. (2020), not Houpert et al. (2018).

For the mid basin section we generate WB1/2 by using potential temperature and absolute salinity values from WB1 above 1600m, and from WB2 below 1600m. We justify this given that the isopycnals are near horizontal between WB1 and WB2 meaning the transport here would not be captured by baroclinic shear. Below 1600 the meridional velocity values of WB2 current meters are used to fill in the region east of WB1/2. The flow over this small area is weak and contributes negligible to transport (Fraser et al., 2020). We are editing the text for clarity.

L. 99:

What is meant by correcting the velocity measurements for "sound"? The instruments are sonic current meters, but what corrections need to be made?

Thank you for pointing out the typo. The current meters are corrected for speed of sound using actual measurements from the nearby hydrographic mooring data. We are editing the text accordingly.

Ll. 107 ff.:

Clarify whether the RTADCP will be used, or why it is worth mentioning here. Else, remove.

Thank you for highlighting. We use the RTADCP to bias-correct the GLORYS12V1 output. The bias-correction of GLORYS12v1 output is only required in the old methodology. We are editing the data and method sections in the manuscript for clarity.

Ll. 138 ff.:

If the Hilbert EOF analysis is not used, I think this paragraph that refers to it can be removed.

Thank you for pointing this out. Our methodology is based on Brandt et al. (2014, 2016, 2021), who applied Hilbert EOF (HEOF) patterns to better capture vertical displacement of current cores. In our study, we evaluate both EOF and HEOF approaches and obtain similar results. For simplicity, we chose to use standard EOF patterns. We are clarifying this choice and are providing additional explanation in the data and methods section and we are removing the paragraph in the results section.

Ll. 140 ff.:

There is a fundamental inconsistency in the method used here, which needs to be reconciled:

For input parameters, the regression method uses a mix of observational data (from the EB1 mooring) and numerical model data (from the GLORYS analysis at a single nearby but separate location). The observational and numerical data reflect two different "realities" that may be inconsistent with each other. There is no quantitative reasoning supporting the choice made here over other available choices, other than the choice of using only observational data from EB1 (which is shown to be inferior). In order to justify the choice made, I recommend additional analyses:

- Validate that the EOF patterns and magnitudes in the model world reasonably match the observational ones, i.e. recreate figure 3 from GLORYS data alone.

- Validate the model against the existing EB1 and glider observations.

- Use the model transport alone (at full model resolution in space and time), and quantitatively test whether this is inferior to your choice.

- Try to find some combination of input from the model (other than the velocities at the single RTADCP location, but simpler than using everything) that might optimize agreement with your reference data.

Thank you for your suggestions. The use of GLORYS12V1 output at the RTADCP location for estimating transport in the eastern wedge was thoroughly tested by Houpert et al. (2020) and calibrated against eight months of ADCP observations at that site, which are not available elsewhere along the section. For the full eastern wedge, they reported a bias error of 0.21 Sv and an RMSE of 0.59 Sv. As noted earlier, our methodology is designed to minimise reliance on model data, using it only where observational gaps cannot otherwise be filled.

Given the strong agreement between glider observations and the previous reconstruction method (Figures 1a; see also Figure 1, page 3 in our response to Reviewer 1), which are independent data products, together with the findings of Houpert et al. (2020) and Fraser et al. (2022), we consider the choice to use GLORYS12V1 output at the RTADCP location justified. As shown in Figure 2b and 2c, the new methodology still depends on this model input to achieve a realistic representation of ESC transport.

[Figure]

*Figure 1 Linear regression of mooring-based reconstructions versus glider-derived transports. Panels show (a) the old methodology (Fraser et al 2022; Houpert et al. 2020), (b) the new methodology using EB1 observations and GLORYS12V1 output at RTADCP, and (c) the new methodology using only EB1 observations. Statistics shown: R = Pearson correlation coefficient; RMSE = Root Mean Square Error; STDE = Standard Error; p = two-sided p-value for the null hypothesis that the slope equals zero.*

Ll. 155 ff.:

I am confused - you are only using the first EOF mode to reconstruct the velocity field, correct? Why does your figure 4b not look like figure 3 (mode 1) then? Shouldn't they be more similar?

Thank you for highlighting a lag of clarity here. It is common practice to remove the temporal mean before performing an EOF analysis as we are interested in the spatial pattern of its variability and not the mean field itself. The EOF pattern shows the dominant mode of variability. Before the regression we remove the temporal mean from the EB1 and GLORYIS12v1 data, reconstruct the section based on the EOF pattern and add the mean of the glider section to the reconstructed velocity anomalies. We are editing the method description for clarity.

Ll. 177-178:

I understand that the mid basin transport is not the primary focus of this study, but since you are mentioning it here and in figures 6-7: The way I read the reference, the mid basin transport is calculated using an assumption of no motion at ~1800 m. This is not consistent with how I read figure 1(b), in that there is more "red" at depth than "blue". I can only assume that there is a substantial amount of variability at that depth. In order to quantify the error from this reference level assumption, can you provide the time series of the velocity (averaged between WB2 and EB1) at 1800 m from the 17 ADCP sections from figure 1(b)? Multiplying this with the water depth and the section width will show you the error in terms of volume transport. I would not be surprised if that error were as large as your entire signal in figure 5. If I am doing the math right, 5 mm/s velocity variability will translate to 1 Sv error, but please double-check and provide the actual number.

Thank you for highlighting the importance of the reference level in the mid basin transport calculation. We agree that the choice of reference level significantly affects the transport estimates. We adopt the method from Houpert et al. (2020). This method yields a basin-wide transport below 1,250 m of approximately −0.3 Sv, aligning with prior findings that deep northward flow is blocked by topography (Holliday et al., 2000), allowing only a small net southward transport of dense Wyville Thomson Overflow Water (-0.3 Sv, Johnson et al., 2017). For the area below the reference level of 1760m, a current of 5mm/s translates to an error of 0.2 Sv. As noted above, Houpert et al. (2020) estimated a total RMSE of 0.96 Sv for the mid basin transport, accounting for both instrumental and methodological uncertainties. We are adding additional information in the data and method section.

Ll. 184 ff.:

I think there is a flaw in how the heat and freshwater transports are derived, in that the underlying temperature and salinity observations do not provide data at locations where it is needed. The way I understand the explanations below equations 4 and 5, the temperature and salinity profiles are from the moorings at the western and eastern edges (and an average of these). However, at least for the mid basin transports, the section is much longer than typical mesoscale length scales (a case made obvious by figure 1). In order for equations 4 and 5 to hold, both the velocity field v as well as Theta and S profiles must be known at some sort of eddy-resolving resolution in situ. I would not have a lot of confidence in the outcome of these equations unless the methodology has been validated somehow (e.g. in a high-resolution numerical model that reasonably depicts the mesoscale eddy field inside Rockall Trough, where you can then compute the ground truth from equations 4 and 5 using the full model field and compare it with a version that resembles sparse observations). If this has

already been done in one of the references, it should be explained here, together with some quantitative uncertainty estimate.

Thank you for highlighting. Please see answer to main comment above.

Ll. 188 ff., eqn. (5):

I think the sign of the salinities is wrong. As it stands, high salinity would give you higher Qf.

Thank you for highlighting. We are correcting the typo.

Ll. 205-207:

If you discuss the undercurrent here, you should refer to it in figure 4 and also include a panel in figure 4 that shows the glider data for comparison. Reg. the overestimate: The error bars overlap, so you could also say that the measurements agree within the given uncertainty, couldn't you?

Thank you for your suggestions. We are updating Figure 3 to include the mean glider section alongside the EOF patterns for direct comparison. For the revised and extended transport calculations (April 2020 – Feb 2023), the mean transport for the glider and the new eastern wedge reconstruction is 1.0±0.3 Sv, while the old methodology gives 1.5±0.2 Sv. The uncertainties overlap, so although the agreement is marginal, the estimates agree within their respective uncertainties. We are editing the text accordingly.

Ll. 208 ff.:

There needs to be an explanation of the bias correction, why it is needed, and what it improves. Perhaps a reference to a publication plus a one-sentence summary is sufficient, but the way it stands, it sounds as if the bias "correction" actually makes agreement with observations here worse.

Thank you for this suggestion. We are moving the explanation of the bias correction in section 3.2 to the data and method section and are editing the paragraph for clarity.

Ll.                                                                                              257-258:

The sentence here uses the words "better" and "robust". I don't think the manuscript in its present state actually demonstrates that the new data are "better", although there are good reasons to believe this is true. It should, however, be demonstrated. As for "robust", I am not sure what that is supposed to mean - it refers to the seasonal cycle, but figure 5 shows that one can basically draw a straight line through the plot at about 1 Sv and be within all the error bars. If anything, this shows that the data cannot determine the presence of an annual cycle with certainty, doesn't it?

Thank you for your comment. As noted in our response to Reviewer 1, we see the following main advantages of the revised methodology:

- Improved accuracy of the mean strength and structure of the ESC, based on multiyear glider observations rather than a bias correction of GLORYS12v1 data using only eight months of ADCP measurements.

- Enhanced ability to reproduce extreme events, likely associated with mesoscale variability, through inclusion of the first two EOF modes.

- High-resolution ESC product in both space and time, reducing aliasing effects caused by temporally scattered glider data.

We are editing the text throughout the manuscript to clarify these points and ensure that the discussion section clearly communicates the value of this methodology.

Ll. 269-271:

This sentence claims that the ESC is "disproportionately important" for something. When I look at figure 6, I find this to be completely untrue on two accounts: One, the heat and freshwater contributions are fairly proportional to the volume transport, and two, the EW contribution is not very important (instead, the total is dominated by the mid basin transport).

Thank you for pointing this out. As noted in our response to Reviewer 1, this statement refers to findings from previous studies (Clark et al., 2022; Daly et al., 2024), which emphasize the importance of the ESC for on-shelf heat and freshwater transports. The term "disproportionately" reflects its relatively small transport compared to the NAC in the Rockall Trough; however, NAC transport in this region does not significantly affect European shelf exchanges, whereas the ESC does. We are revising the sentence for clarity.

**Improvements to text (readability/appearance/typos):**

Abbreviations and acronyms:

The amount of abbreviations is overwhelming. Some are not used consistently throughout the text (e.g. RTWB1 vs. WB1). Some are never spelled out (geogr. names from fig. 1). To make things a bit easier, be sure to:

- spell out each abbreviation at its first occurrence and additionally in each figure caption if it occurs inside a figure,

- avoid inconsistencies,

- consider adding a list of all abbreviations in the supplemental materials.

Thank you for highlighting this point. We are revising the text and figures to ensure that abbreviations and acronyms are used consistently throughout the manuscript. Each abbreviation is now spelled out at its first occurrence and in figure captions where applicable. In addition, we are considering adding a comprehensive list of abbreviations in the supplementary materials for ease of reference.

Ll. 17/18:

Remove "Fu et al." reference if unpublished, else update here and add proper citation to the reference list.

Fu et al. (2025) is now published and we are adding it to the reference list accordingly.

Ll. 61-62:

Remove "PSU".

This is being done.

L. 64: Change "Avaraged" to "Averaged"

This is being done.

L. 105: Change "gab" to "gap"

This is being done.

GLORYS references: When I search for "GLORYS" in the manuscript, I see inconsistent occurrences of GLORYS12, GLORYS21, GLORYS2, followed by equally inconsistent version numbers 1, 2, or 12. Please determine the exact version number used, and correct this throughout the manuscript. Why not just call it "GLORYS" everywhere and reference the full name once and once only with the dataset citation?

This is being done.

Ll. 158-159: I found the reference to the future equations confusing. This sentence can perhaps be removed.

This is being done.

L. 250:

I am wondering if "Conclusions" would be a more appropriate title than "Discussion" here.

We agree with the reviewer and are renaming the section to "Conclusions." For clarity, we are revising the overall paper structure and are adding a dedicated discussion section.

L. 272:

Change "targetted" to "targeted"

This is being done.

**Suggestions for figures and captions:**

Use consistent labels and spelling across figures:

Figures 1(b), 3, 4, 6, S1 all have depth as vertical axis, but sometimes it counts positive up, sometimes down, and the axis label is spelled differently almost every time.

Ditto for 1(a), 1(b), 3, 4, S1 for longitude.

Ditto for time in 2, 7, 8, S2.

Thank you for highlighting this point. We are revising all figures and captions to ensure clarity and consistency throughout the manuscript. This includes harmonizing labels, abbreviations, and formatting across figures and captions.

**Figure 1:**

Add (a) to top panel.

Increase size of panel (b) such that it is roughly as wide as panel (a).

Increase font size of panel (b) such that it roughly matches that of panel (a).

In caption, start the panel (b) part with verbiage that says what is shown (e.g., "cross-section view of Rockall Trough section").

In caption, be consistent with abbreviations - use either RTWB1 or WB1, ditto for ...2.

Panel (b) shows things that are not explained in the caption - explain or remove these: two dashed lines with red dots on top, one solid line with blue dot on top. I assume these are the same as in figure 4, but this needs clarification.

Use the same green symbols on panels (a) and (b), i.e. either circles on both panels, or triangles on both.

Either explain all abbreviations in the caption (include the geographic names of panel (a)), or point to a list (potentially in the supplemental materials) where they are explained.

In caption, change "focusses" to "focuses".

Thank you for your suggestions. We are editing Figure 1 and caption accordingly.

**Figure 2:**

Add cruise IDs to potential list of acronyms, and refer to this list in the caption.

In caption, change "CTD sensors (d-f)." to "CTD sensors (c-f)."

Thank you for your suggestions. We are removing the cruise labels and are editing the caption accordingly.

**Figure 3:**

The figure caption here refers to "meridional velocity", whereas figure 4 mentions "across section velocity". These two things are almost identical, but not 100%. Please confirm that each caption correctly describes the quantity shown, or correct as appropriate.

Make the axes limits identical to those from figure 4, and add the same vertical lines/symbols for orientation.

Thank you for highlighting. We are revising all relevant figures to show meridional velocities and to share the same limits on the vertical axis.

**Figure 4:**

Use the same symbols and colors as figure 1 (a) and (b) for the mooring etc. locations.

This is being done.

**Figure 5:**

The figure seems to show volume transport, but the caption calls the data "velocities depth-averaged…". Correct the caption such that it calls out the correct physical quantity.

Add the time periods for the blue and green curves in the legend (why just the orange and black?).

Thank you for spotting this. We are updating the relevant figure and are clarifying the caption to ensure accuracy (see updated version in Figure 2).

**Figure 8:**

The font size is inconsistently large compared to the other figures. Reduce font size to make it look similar to the others.

This is being done.

---

## Author Comment (AC3)

**Response to Reviewer 4 for "Eight years of continuous Rockall Trough transport observations using moorings and gliders"**

We thank the reviewers for their constructive comments which help us improve the quality of our manuscript. Below, we provide detailed responses to each comment.

During the revision, we are implementing the following improvements to the transport calculations, which we are making independently to strengthen the analysis:

- **Extending the mooring dataset to 10 years**, now overlapping with the entire glider observation period. We are changing the title of the manuscript to reflect this update: "A decade of continuous Rockall Trough transport observations using moorings and gliders"

- **Correcting the EOF analysis** by using the original time steps of glider transects instead of 15-day averages, which previously included an irregular number of transects.

- **Correcting an error in the new methodology:** The EOF analysis and regression are applied to velocity anomalies. In the earlier version, we mistakenly subtracted the glider mean at EB1 and RTADCP positions from EB1 data and GLORYS12V1 output instead of subtracting the mean of each respective dataset. This introduced a systematic offset, which has now been corrected. The glider mean field is added at the final step to define the mean of the reconstructed section, eliminating the need for bias correction of the GLORYS12V1 output.

**Review of the manuscript egusphere-2025-3167.pdf**

**General comments**

The manuscript, entitled "Eight years of continuous Rockall Trough transport observations using moorings and gliders", presents new estimates of the European Slope Current (ESC) and North Atlantic Current (NAC) volume transports through the Rockall Trough (RT), adding two years to the existing time series. For the first time, heat and freshwater transports are also presented. The new data set includes temporal overlap of mooring and glider data from the ESC. The authors have developed a new methodology, where mooring and glider data are combined, allowing for better estimates of the ESC transports. The new methodology gives a better representation of the Eastern Wedge covering the ESC, with the undercurrent now visible in the data. Interestingly, they also find, that the variability of the ESC seems independent from the NAC variability. Over the eight years of observations, the total RT transport does not have a significant trend. The manuscript highlights the benefits of

combining various data sets to get a better representation and understanding of physical processes in the ocean.

Generally, the manuscript is within the scope of OS and the language is fairly good. But the lack of consistency in e.g. figures and abbreviations gives the impression that the manuscript is written in a hurry and that the authors have not spent enough time on polishing the manuscript. Also, the discussion is relatively short and appears more like a summary of the results section, while the results are discussed already in that section.

Thus, major revisions are needed before publication in Ocean Science.

Thank you for your thorough feedback. We appreciate your observations regarding consistency, language, and the need for a more comprehensive discussion. In response, we are undertaking major revisions to the manuscript. Specifically, we are reviewing and refining the structure, text, figures, and captions to improve clarity, completeness, and consistency. Additionally, the discussion section is being substantially rewritten to provide a deeper interpretation of the results rather than a summary

**Specific comments**

**Abstract**

Line 8-9: Here you say, that you have produced, for the first time, a continuous ESC transport time series, but as I understand, the volume transport is an update of Fraser et al., 2022. On the other hand, heat and freshwater transports for ESC and RT are presented for the first time. I suggest that you highlight that in the abstract.

Thank you for this helpful observation. You are correct that the volume transport represents an update of Fraser et al. (2022), while the heat and freshwater transports for the ESC and RT are presented for the first time. We are revising the abstract to clearly emphasize this distinction and are ensuring it accurately reflects the full scope of the analysis.

**Introduction**

Line 15-16: The listing of impacted regions could be rearranged in order of appearance downstream of RT, i.e. Arctic last.

Thank you for this suggestion, we are editing the sentence accordingly.

Line 29: "together with"

This is being changed.

**Data and Methods**

In your description, you use both "glider" and "gliders". I suggest being consistent.

Thank you for highlighting. We are editing the text accordingly to use gliders.

Line 59: Cross-hatched region is not visible on a print out. See also comments to Figure 1 below.

Thank you for highlighting. The revised Figure 1 does not include the cross-hatched area anymore.

Line 61: Should it be "**an** SBE41"?

We are editing the sentence to: "All gliders are equipped with SBE41 CTD senors, which…"

Line 91-95: See comments to Figure 1 below.

See answer to Figure 1 comment below.

Line 107-108: This sentence gives the impression that no data exist from the RTADCP. Please modify.

Thank you for pointing this out. The RTADCP was deployed in the ESC core in 2014 but only the first 8 months of data could be recovered. Any other attempt to recover data from later deployed ADCPs failed as the instruments were severely damaged presumably through fishing activities (Houpert et al., 2020). The old Rockall Trough transport product hence relies on ocean reanalysis output, which is bias-corrected by the available 8-month RTADCP time series, to reconstruct the northward flow of the ESC following (Houper et al., 2020; Fraser et al., 2022). We are adding this information to the data and method section for clarity.

Line 159: Please add, that Eq. 3-5 are given in section 3.4.

Thank you for this suggestion. We are removing the sentence and are incorporating the relevant information directly after introducing the equations to improve clarity.

Line 185 and 189: How did you select the reference values? Please clarify in the text.

Thank you for pointing this out. As mentioned in our responses to Reviewer 3 (answer to L185, page 5), the reference temperature and salinity are defined as the mean values observed at WB1 and WB2, which are located in the southward-flowing current west of the NAC (Figure 1b in manuscript). This choice sets the mean heat and freshwater transport through the western wedge to zero, so that the transports calculated for the mid basin and eastern wedge primarily reflect the signals of the NAC and ESC, respectively. We are including this information in the manuscript for clarity.

**Results**

I suggest that you add a table listing the various transport values given in the text and maybe also correlation coefficients. A table gives a better overview and it is easier for the reader to compare the different branches.

Thank you for this suggestion, we are adding tables with the important transport statistics in the result section where appropriate.

Line 200: Please specify Figure 5a.

This is being done.

Line 203: Please add Sv to the transport values.

For the revised and extended transport calculations (April 2020 – Feb 2023), the mean transport for the glider and the new eastern wedge reconstruction is 1.0±0.3 Sv, while the old methodology gives 1.5±0.2 Sv. The uncertainties overlap, so although the agreement is marginal, the estimates agree within their respective uncertainties. We are editing the paragraph accordingly.

Line 224: Replace "small" with "close to zero"

See answer to lines 185 and 189

Line 225: "**an** increasing trend"? Are these trends significant?

We are updating the trends according to the revised and extended transport time series. We are including a table in the supplementary information that reports these trends along with their significance. All trends for the western wedge and the mid basin are significant, except for the freshwater transport in the mid basin, and we do not find significant trends in the eastern wedge transports.

**Discussion**

As mentioned above, the discussion is relatively short and is more like a summary of the Results section. Must be rewritten into a proper discussion section.

Thank you for this suggestion. We agree and are restructuring the manuscript accordingly. Specifically, we are changing the title of the section to "Conclusion" and are adding an additional discussion section.

Line 265: "controlled by"

This is being changed.

**Figures and legends**

Figure 1: This figure needs some updates. Firstly, the resolution in both panels is too low.

The map must be marked with an a). The green circles are not very visible and it is hard to see, that there actually are three of them. The speed on the map seems okay, but what are the arrows based on? I do not have comments on the swirls here and there, but the arrow located at ~12°W; ~64°N is flowing in the wrong direction! See e.g. Hansen et al., (2023, Fig 14) or Orvik and Niiler (2002, Fig 1 and 3b).

Hansen et al., 2023 (https://doi.org/10.5194/os-19-1225-2023); Orvik and Niiler 2002 (https://doi.org/10.1029/2002GL015002)

The hydrographic section in b) is from Frazer et al. 2022 (their Fig 1.b in a good resolution), but the resolution here is too low to see details in the figure. Houpert et al., 2020 have a similar figure (their Fig 2.a) and they nicely define the WW, MB and EB areas on top of the section. Together with the limit at depth, this would give a much better illustration of which area the three transport estimates are calculated for. As the figure is now, the cross-hatching is hard to see, especially on print, and the different symbols on the surface are not described. Please update the figure and legend accordingly.

Thank you for your suggestions. We are revising Figure 1 and its caption to enhance clarity, completeness, and consistency, and we are correcting the representation of the sketch currents accordingly.

Figure 2: Fine resolution. The names on the top are cruise id's? Please clarify in the legend.

Thank you for highlighting this point. We are revising Figure 2 to ensure consistency with the other figures and are removing the cruise names for improved clarity.

Figure 3: Please use "°W" for consistency with other figures.

Thank you for your suggestion. We are revising Figure 3 accordingly to ensure consistency with the other figures

Figure 4: Blue square marks the RTADCP – right? This information should also be added to Figure 1b.

Thank you for pointing this out. We are revising Figure 4 to be consistent with Figure 1.

Figure 5: The figure legend is insufficient. Reference to a) and b) is missing. The black dotted lines and the vertical bars (in b) are not described in the legend. Moving on to figures 6, 7 and 8, you here use the colors blue, orange and green for MB, EW and WW – but they are not consistent in all figures. Please select one color for each region and use them consistently.

Please do not re-use these colors in Figure 5 (except for the EW color). What is the temporal resolution in a)?

Thank you for highlighting. We are revising Figure 5 and caption for clarity and consistency (see Figure 1 in our response to Reviewer1, page 3).

Figure 6: Please repeat the description of the numbers in the panels.

Thank you for highlighting this point. We are revising the caption of Figure 6 to ensure completeness and clarity.

Figure 7: Please add more details to the legend and include ref to all panels. The red lines (solid and dotted) in b, d and f are not very visible. Please modify.

Thank you for highlighting this point. We are revising the Figure 7 and caption to ensure completeness and clarity.

Figure 8: Please be consistent in the use of colors and avoid to use orange for temperature in a) and blue for WB1/2 in b). You should probably also avoid to use red and green in the same figure.

2$^{nd}$ line in legend should read "(b) isolating".

Thank you for your suggestion. We are revising the Figure 8 and caption to ensure completeness and clarity.

**Supplementary**

The title is different from the manuscript title.

Thank you for highlighting. We are editing the title accordingly.

---

## Author Comment (AC5)

**Response to Reviewer 3 for "Eight years of continuous Rockall Trough transport observations using moorings and gliders"**

We thank the reviewers for their constructive comments which help us improve the quality of our manuscript. Below, we provide detailed responses to each comment.

During the revision, we are implementing the following improvements to the transport calculations, which we are making independently to strengthen the analysis:

- **Extending the mooring dataset to 10 years**, now overlapping with the entire glider observation period. We are changing the title of the manuscript to reflect this update: "A decade of continuous Rockall Trough transport observations using moorings and gliders"

- **Correcting the EOF analysis** by using the original time steps of glider transects instead of 15-day averages, which previously included an irregular number of transects.

- **Correcting an error in the new methodology:** The EOF analysis and regression are applied to velocity anomalies. In the earlier version, we mistakenly subtracted the glider mean at EB1 and RTADCP positions from EB1 data and GLORYS12V1 output instead of subtracting the mean of each respective dataset. This introduced a systematic offset, which has now been corrected. The glider mean field is added at the final step to define the mean of the reconstructed section, eliminating the need for bias correction of the GLORYS12V1 output.

The paper presents an 8-year time series of volume, heat, and freshwater transports through the Rockall Trough in the eastern subpolar North Atlantic. The estimates utilize glider and mooring measurements, together with data from an ocean reanalysis product. These transports are critical in inducing hydrographic changes downstream within the subpolar gyre as well as further north in the Nordic Seas, which makes accurate estimates highly desirable. In particular, this analysis emphasizes a new method that it introduces for the wedge transport estimates, along with the new freshwater/heat transport estimates. However, I have two main concerns about both: it is hard to see a clear improvement from the new method compared to the old method, so it would require thorough validation; the seemingly arbitrary definitions of freshwater/heat transport would limit the usefulness of the estimates, e.g., when comparing them to other existing estimates in the region. I recommend a moderate/major revision with more detailed comments outlined as follows.

Main comments:

1. The authors proposed a new method for reconstructing the velocities in the eastern wedge for the upper 1000 meters, which had been previously reconstructed primarily using velocity data from an ocean reanalysis product. However, the robustness of these new glider-based estimates requires careful and more thorough validation. The first main issue is related to the velocity derived from the glider paths. The authors acknowledge this issue (lines 84-85) but unfortunately did not address the associated uncertainty and its subsequent impact. A related issue concerns the comparisons with various other estimates, which are briefly discussed in section 3.1. As this new method is poised to be a key improvement over previous estimates, a thorough discussion and stronger evidence of its superiority are needed.

Thank you for raising this important point regarding the robustness of the transport estimates. We agree that additional clarification is needed and are expanding the manuscript to address uncertainty and validation in detail.

For the old methodology, Houpert et al. (2020) previously assessed errors associated with horizontal extrapolation and reliance on ocean reanalysis at the eastern wedge by comparing against LADCP data from cruises along the section. For the eastern wedge transports, they reported a mean bias of –0.21 Sv and an RMS error of 0.59 Sv (see their SI text S1.3, Table S1, Figures S6 and S8). The glider product used here is an extension of the Fraser et al. (2022) dataset, which has undergone peer-reviewed scrutiny. In the revised analysis, the glider and mooring observation periods now overlap, allowing direct comparison. As shown in the manuscript (Figure 5) and in the extended time series (refer to Figure 1 in our responses: Reviewer 1 on page 3 and Reviewer 2 on page 9), the glider-derived transports agree well with those obtained using the previous methodology. This consistency across independent data products supports the robustness of both approaches.

As noted in our responses to Reviewer 1 and 2, we see the following main advantages of the revised new methodology:

- Improved accuracy of the mean strength and structure of the ESC, based on multiyear glider observations rather than a bias correction of GLORYS12v1 data using only eight months of ADCP measurements.

- Enhanced ability to reproduce extreme events, likely associated with mesoscale variability, through inclusion of the first two EOF modes.

- High-resolution ESC product in both space and time, reducing aliasing effects caused by temporally scattered glider data.

We are editing the text throughout the manuscript to clarify these points and ensure that the discussion section clearly communicates the value of this methodology.

2.The overall results and the analysis are comparatively limited. Much of the focus is on the volume transport and is similar to the previous publication based on shorter records. The authors indicated that the heat and freshwater transport estimates are presented for the first time here; if so, I believe they deserve a more in-depth analysis. Some questions could be readily addressed by the data, e.g., what is the relative contribution of velocity versus T/S variations to the transport variability? Which subregion dominates the T/S transport variability? Caution is needed regarding the definition of the heat and freshwater transports, as this may introduce uncertainties and make it difficult to interpret the results when comparing them to other estimates in the region. I would suggest the authors clarify what they are presenting and include sufficient information on how to contextualize the presented estimates with those in previous publications.

Thank you for highlighting the need for a more comprehensive treatment of heat and freshwater transport. We agree that the initial presentation lacked sufficient detail and are revising the manuscript accordingly.

First, we are expanding the data and methods section to clarify our approach. Specifically, we are now including an assessment of using mooring-based hydrographic profiles by comparing transports derived from full ship and glider sections with those calculated using profiles subsampled at mooring positions. This comparison is being detailed in the revised manuscript and summarized in the response to Reviewer 2 (page 4).

In addition, we are extending the analysis of heat and freshwater transports to address the reviewer's suggestions:

- Quantify the relative contributions of velocity versus temperature/salinity variations to heat and freshwater transport variability.

- Identify which subregions dominate variability.

- Discuss the sensitivity of the estimates to the chosen reference values for temperature and salinity, and how these choices influence interpretation.

- Provide context by comparing our estimates with those from previous studies in the region, highlighting similarities and differences.

These additions aim to improve clarity, reduce uncertainty, and ensure that the presented results can be meaningfully interpreted and compared within the broader framework of subpolar North Atlantic transport studies.

Other comments:

Line 10: This is a commendable goal, but the framework's validity requires more thorough demonstration.

Thank you for pointing this out. Please see answer to first main comment.

Line 77: Add more details on the common section?

Thank you for highlighting. We are revising the section for clarity.

Figure 1: Please label all moorings in the main and inset plots. For the caption: is 'RTWB1' the same as 'WB1'? Same question for 'RTWB2' and 'WB2'.

Thank you for highlighting. We are revising Figure 1 and caption for clarity and consistency.

Figure 2 and the related text: The notation 'WB 1/2' is confusing. Please clarify if it referes to a single mooring location or a composite of WB1 and WB2.

Thank you for pointing this out. We are revising Figure 2 and the related text to improve clarity. In the updated manuscript, WB1/2 is explicitly defined as a composite of WB1 and WB2. Additional details are provided in our response to Reviewer 2 (page 6, comment on lines 93–95)

Section 2.3: Please elaborate on why the supplementary datasets are necessary and how they are specifically used in the analysis.

Thank you for pointing out the lag of clarity for what the auxiliary dataset are use. We are editing the section accordingly.

Line 117: Please provide clear definition for 'western wedge', 'mid basin' and 'eastern wedge'. Their spatial extents are not clear.

Thank you for highlighting. We are editing Figure 1b, to show the extent of the different sections. The western wedge extends between 12.5°W to 13°W, the mid basin extends between 12.5°W and 9.6°W and the eastern wedge extends between 9.6°W and 9.2°W over the upper 1760m.

Line 140: Please label 'RTADCP' in the relevant figures.

This is being done.

Line 151: What is 'glider data at EB1'? If velocity data from gliders are used at EB1, I would suggest additional tests to validate the reconstruction. For instance, compare a reconstruction using: (a) mooring velocity at EB1 + first EOF, (b) velocity from GLORYS at the ADCP location + first EOF. These two tests would better quantify importance of the EB1 and ADCP velocities for the eastern wedge transport.

Thank you for highlighting this point and for suggesting additional validation tests. To clarify, we subsampled the glider data at the positions of EB1 and RTADCP to verify the reconstruction approach. Following your recommendation, we have revised the methodology to strengthen the comparison. Specifically, we are now averaging the mooring data and GLORYS output within ±1 day of each glider time step, ensuring consistency in temporal alignment. Details of this revision are provided in Figure 1 of our response to Reviewer 2 (page 9).

Line 160 (Figure 4): To directly assess the impact from the new method, please include a comparison of the transport time series from the new and old methods.

Thank you for this suggestion. We confirm that the comparison between the old and new methodologies is already included in the manuscript. Specifically, both transport time series are shown in Figure 5a in the manuscript, and additional comparisons are provided in Figures 4 and 6 as well as in Section 4.2 in the manuscript.

Line 174: The description of the velocity reconstruction is confusing. Please elaborate with more details. Specifically, clarify what 'WB1/2' refers to and where the 'WB1/2 position' is, both in the text and on the figures.

Thank you for highlighting this point. WB1/2 refers to the midpoint between WB1 and WB2. We are revising the text for clarity and are adding labels in Figure 1b to clearly indicate its position.

Line 185: How was the reference temperature determined? Elaborate on how to the choice of the reference influences the interpretation of the resulting temperature transport.

The reference temperature and salinity are defined as the mean values observed at WB1 and WB2, which are located in the southward-flowing current west of the NAC (Figure 1b in manuscript). This choice sets the mean heat and freshwater transport through the western wedge to zero, so that the transports calculated for the mid basin and eastern wedge primarily reflect the signals of the NAC and ESC, respectively. We are adding this information to the data and method section and are discussing the choice in the discussion section.

Line 189: Similar for the choice of salinity reference (Sref). A discussion on the sensitivity of the freshwater transport results to different Sref values is needed to understand the robustness of the estimates.

See answer to comment above.

Line 219: The reconstruction relied on a reanalysis product in addition to the mooring data. Please clarify.

Thank you for this comment. To clarify, our comparison of total volume transport through the Rockall Trough includes previous reconstructions for shorter time periods (Houpert et al., 2020; Fraser et al., 2022). All three estimates rely on the same mooring data and GLORYS reanalysis output. We did not introduce an additional total transport product, so we believe the current description is sufficient.

Figure 5: The labels are confusing. Please specify what each line represents in the caption. Also, provide more details on how the estimates were obtained. For example, what is the difference between 'old' and 'old resampled'?

Thank you for highlighting this point. We are revising Figure 5 and caption to improve clarity (see Figure 1 in our response to Reviewer 1, page 3).

Lines 203-204: Be cautious with phrases like 'differ notably'. Specify what the numbers after plus/minus represent (e.g., standard errors). And discuss the statistical significance of the mean differences.

Thank you for highlighting. For the revised and extended transport calculations (April 2020 – Feb 2023), the mean transport for the glider and the new eastern wedge reconstruction is 1.0±0.3 Sv, while the old methodology gives 1.5±0.2 Sv. The uncertainties overlap, so although the agreement is marginal, the estimates agree within their respective uncertainties. We are editing the text accordingly.

Line 206: The analysis should consider the error bars (uncertainty) for both estimates when comparing them.

See answer to comment above.

Line 212: How does the gliders' irregular sampling affect the mean transport estimate? The glider-based transport is used as a benchmark in section 4.1, but its own robustness should be evaluated more thoroughly earlier in the paper.

Thank you for raising this important point. With a total of 166 glider transects over a three-year period and strong agreement with independent estimates from the previous methodology, we consider the mean transport estimate derived from glider data to be

robust. We recognize, however, that this is a critical aspect of our analysis and are adding a paragraph in the discussion section to explicitly address the implications of irregular glider sampling on transport estimates, including a comparison with mean EW values from previous studies (1.3 ± 0.2 Sv in Fraser et al., 2022 and 1.4 ± 0.3 Sv in Houpert et al., 2020).

Figure 6: The black line is also dashed – does it represent the old method? Please clarify in the caption.

Thank you for pointing this out. We are updating Figure 6 to use consistent solid lines for all transports derived using the new methodology.

Line 223: The heat and freshwater transport values are highly sensitive to the arbitrary choice of the respective reference, especially given the non-zero volume transport across the section. This major caveat should be emphasized.

Thank you for pointing this out. We are emphasising this in our discussion.

Figure 7bdf: what do the vertical color bars represent? The mean value appears to be incorrectly placed in panel f – please verify.

Thank you for highlighting this point. In panels (b) and (d), red shading marks periods when the filtered time series exceeds the mean plus one standard deviation, and blue shading marks periods when it falls below the mean minus one standard deviation. In panel (f), the shading is reversed. We are editing the caption for clarity and confirmed that the mean value text is positioned to avoid overlapping with the time series and improve readability.

Line 240: The analysis would be significantly enhanced by investigating the potential causes of the observed differences. For example, what is the relative contribution by T/S vs velocity changes? How is it related to volume transport and T/S properties in the specific wedges and mid basin?

Thank you for these suggestions. We agree and are incorporating additional analysis as outlined in our response to the second main comment above.

Line 248: How do property changes in the region subsequently affect the corresponding transports across the Trough? This would provide a strong linkage to the broader analysis.

Thank you for pointing this out. Figure 8 illustrates how temperature and salinity influence transport estimates in the mid basin, which dominates the total transport variability. We acknowledge that the manuscript currently includes only a brief discussion of this relationship and are enhancing the results section with additional detail to strengthen the linkage to the broader analysis.

Line 253: It would be valuable to contextualize these transport estimates within the broader understanding of subpolar volume, heat, and freshwater transports.

Thank you for this suggestion. We agree and are revising the paper structure to include a new discussion section that places our transport estimates within the broader context of subpolar volume, heat, and freshwater transports. In this section, we are comparing our results with those from Dotto et al. (2025), which focus on NAC branches west of the Rockall–Hatton Bank, and with OSNAP observations (e.g., Fu et al., 2025) to highlight similarities and differences. We are also discussing our findings in relation to broader subpolar variability, including the subpolar gyre and dominant atmospheric modes of variability.

Line 256: While the new method is a key novelty, its necessity and efficacy have not been thoroughly validated. A more rigorous comparison with the old method and a clear demonstration of its improvement are necessary.

Please see answer to first main comment.

Line 267: It would be great to include an analysis for assessing the effect of salinity changes.

Thank you for this suggestion. We agree and are extending the result section to include an analysis of salinity changes, as shown in Figure 8 and S2.

Line 270: Again, my suggestion is to separate and quantify the impact of volume transport variability in different parts of the section on the overall property transport variability.

Thank you for these repeated suggestions. We agree and are addressing them as outlined in our response to the second main comment above.

Line 272: Consider adding a discussion on how the presented results are related to the other transport estimates within the subpolar region.

See response to comment of line 253.